# BRCA2 antagonizes classical and alternative nonhomologous end-joining to prevent gross genomic instability

Jinhua Han[1], Chunyan Ruan[1], Michael S.Y. Huen[2], Jiadong Wang[3], Anyong Xie[4], Chun Fu[5], Ting Liu[6] & Jun Huang [1]

BRCA2-deficient cells exhibit gross genomic instability, but the underlying mechanisms are not fully understood. Here we report that inactivation of BRCA2 but not RAD51 destabilizes RPA-coated single-stranded DNA (ssDNA) structures at resected DNA double-strand breaks (DSBs) and greatly enhances the frequency of nuclear fragmentation following cell exposure to DNA damage. Importantly, these BRCA2-associated deficits are fueled by the aberrant activation of classical (c)- and alternative (alt)- nonhomologous end-joining (NHEJ), and rely on the well-defined DNA damage signaling pathway involving the pro-c-NHEJ factor 53BP1 and its downstream effector RIF1. We further show that the 53BP1–RIF1 axis promotes toxic end-joining events via the retention of Artemis at DNA damage sites. Accordingly, loss of 53BP1, RIF1, or Artemis prolongs the stability of RPA-coated DSB intermediates in BRCA2-deficient cells and restores nuclear integrity. We propose that BRCA2 antagonizes 53BP1, RIF1, and Artemis-dependent c-NHEJ and alt-NHEJ to prevent gross genomic instability in a RAD51-independent manner.

[1] Life Sciences Institute and Innovation Center for Cell Signaling Network, Zhejiang University, Hangzhou, Zhejiang 310058, China. [2] School of Biomedical Sciences, Li Ka Shing Faculty of Medicine, The University of Hong Kong, Pok Fu Lam, Hong Kong, China. [3] Institute of Systems Biomedicine, Department of Radiation Medicine, School of Basic Medical Sciences, Peking University Health Science Center, Beijing 100191, China. [4] Institute of Translational Medicine, College of Medicine, Zhejiang University, Hangzhou, Zhejiang 310058, China. [5] Department of Obstetrics and Gynecology, The Second Xiangya Hospital of Central South University, Changsha, Hunan 410011, China. [6] Department of Cell Biology, Zhejiang University School of Medicine, Hangzhou, Zhejiang 310058, China. Jinhua Han and Chunyan Ruan contributed equally to this work. Correspondence and requests for materials should be addressed to J.Hu. (email: jhuang@zju.edu.cn)

**D**NA double-strand breaks (DSBs) are the most deleterious form of DNA damage, which if left unrepaired or mis-repaired, could lead to chromosomal aberrations and cell death[1]. To counteract the deleterious effects of DSBs, cells have evolved two major DSB repair pathways—classical non-homologous end-joining (c-NHEJ) and homologous recombination (HR), both of which are highly conserved from yeast to humans[2,3]. C-NHEJ is a relatively fast and efficient process that involves direct ligation of the two broken DNA ends, and has been shown to be active throughout interphase[4]. The key components of c-NHEJ include the DNA end-binding heterodimer Ku70/80, the kinase DNA-PKcs, the nuclease Artemis, the DNA ligase IV, the scaffolding proteins XRCC4 and XLF, and the newly characterized PAXX[4,5].

In contrast to NHEJ, HR operates with slower kinetics and is executed primarily in the late S and G2 phases of the cell cycle when sister chromatids are available as repair templates[2,3]. HR is initiated by the 5′ to 3′ nucleolytic resection of DSB ends, a process mediated by the MRE11–RAD50–NBS1/XRS2 (MRN/X) complex in conjunction with CtIP/Sae2 that carries out limited resection, and the 5′–3′ exonuclease EXO1 or the helicase–nuclease protein complex BLM/Sgs1-DNA2 that carries out extensive resection[6,7]. The resulting 3′ single-stranded DNA (ssDNA) overhangs are rapidly coated by replication protein A (RPA) to prevent the formation of secondary structures such as hairpins[8]. In the subsequent step, the recombinase RAD51 replaces RPA, with the help of recombination mediator proteins, to form RAD51 nucleofilaments[2,9–11]. These nucleofilaments then catalyze homology search, followed by DNA strand invasion, DNA synthesis, and ligation of the recombinant products.

In addition to c-NHEJ and HR, at least two other modes of DSB repair, namely single-strand annealing (SSA) and alternative nonhomologous end-joining (alt-NHEJ), have been described in both normal and pathological contexts[12,13]. SSA specifically occurs when a DSB is induced between two stretches of repetitive sequence oriented in the same direction[13,14]. Similar to HR, SSA requires extensive DNA end resection[13,15]. Once a homology sequence is exposed in the 3′ overhangs, RAD52, the central protein in SSA, catalyzes the annealing of complementary ssDNA[13,16]. Subsequently, the sequences between the repeats are cleaved off by the ERCC1–XPF endonuclease complex and the resulting gaps are filled by DNA polymerase and sealed by DNA ligase[13]. It is noteworthy that SSA does not require a strand invasion step and thus is genetically independent of RAD51[13].

Alt-NHEJ was originally identified as a backup pathway to repair DSBs when c-NHEJ is disabled[12,15,17–25]. However, emerging evidence demonstrates that alt-NHEJ can also occur in c-NHEJ-proficient cells[12,20,26]. Alt-NHEJ demands PARP1-dependent DSB synapsis and relies on DSB end processing by the MRN/X-CtIP/Sae2 protein complex to expose micro-homology which allows annealing of broken DSB ends[20,27–32]. After removing the overhanging noncomplementary 3′ flaps, the flanking single-stranded regions created on both strands through resection are filled in by the low-fidelity DNA polymerase theta (Polθ), and the remaining nicks at alt-NHEJ sites are ligated primarily by DNA ligase 1 and DNA ligase 3[24,27,33–36]. Remarkably, in addition to its role in the fill-in synthesis process, Polθ has also been shown to promote DNA synapse formation and strand annealing[33]. Although considerable progress has been made recently toward understanding the operational framework of alt-NHEJ in mammalian cells, identity of the DNA nuclease(s) required for the removal of 3′ flaps from the annealed intermediate remains to be defined.

*BRCA2* is a tumor suppressor gene in which its germline mutations predispose individuals to early development of breast and ovarian cancers[37]. Cells deficient in BRCA2 are hypersensitive to DNA damaging agents and exhibit gross genomic instability[37]. The best-known attribute of the BRCA2 protein is its involvement in HR repair, where it mediates RAD51 nucleation onto ssDNA structures[11,38]. Although it has been speculated that the error-prone DSB repair pathways such as c-NHEJ, alt-NHEJ, and SSA may contribute to the genomic instability phenotype observed in cells lacking BRCA2, their relative contributions remain largely unknown. In addition to its essential role in HR, recent studies have revealed a HR- and DSB-independent function for BRCA2 during replication stress[39–43]. Specifically, BRCA2 protects nascent DNA strands from PTIP- and Mre11-dependent degradation at stalled replication forks[39–43]. Similar to its role in HR, the function of BRCA2 in fork protection also depends on its interaction with PALB2[44].

In this study, we found that inactivation of BRCA2 but not RAD51 triggered premature dissolution of damage-induced RPA2 foci, and led to elevated incidence of nuclear fragmentation events. Genetic ablation of key components of both the c-NHEJ and alt-NHEJ pathways suppressed these aberrant phenotypes in BRCA2-depleted cells. We further uncover that the increased aberrant c-NHEJ and alt-NHEJ activities associated with BRCA2 loss are largely dependent on the pro-c-NHEJ factor 53BP1 and its downstream effectors RIF1 and Artemis. Our results indicate that the 53BP1–RIF1–Artemis axis drives gross genomic instability in BRCA2-deficient cells by promoting aberrant activation of c-NHEJ and alt-NHEJ.

## Results

**BRCA2 inactivation accelerates the dissolution of RPA2 foci.** BRCA2 mediates the replacement of RPA with RAD51 on single-stranded DNA (ssDNA) molecules to enforce HR-dependent DSB repair. To better understand the role of BRCA2 in faithful DSB repair, we analyzed the kinetics of IR-induced RPA2 foci formation in control and in BRCA2-depleted cells. As shown in Fig. 1a–c and Supplementary Fig. 1a, in response to IR, RPA2 foci formation in control cells increased to ~30% at 1 h, peaked at 24 h (around 80%), and then gradually declined. Notably, while depletion of BRCA2 impaired RAD51 foci formation as previously described, quantification of RPA2 foci at 1, 2, 6, or 12 h after IR treatment did not reveal a requirement of BRCA2 (Fig. 1a–c and Supplementary Fig. 1a), suggesting that the tumor suppressor protein is not essential for the formation of RPA2 foci. Surprisingly, IR-induced RPA2 foci consistently resolved with much faster kinetics in the absence of BRCA2 (Fig. 1a–c and Supplementary Fig. 1a). Similar results were obtained when cells were treated with the Topoisomerase I inhibitor CPT (Supplementary Fig. 1b, c). We further examined the impact of BRCA2 deficiency on the kinetics of RPA2 foci dissolution in BRCA2-deficient VC-8 Chinese hamster ovary cell line and a derivative that has been reconstituted with wild-type BRCA2. As shown in

**Fig. 1** BRCA2 depletion accelerates the dissolution of RPA2 foci and causes massive nuclear fragmentation. **a** HeLa cells were transfected with control siRNA or siRNAs against BRCA2. Forty-eight hours post transfection, cells were exposed to 10 Gy IR and then allowed to recover for the indicated time periods before being processed for immunofluorescence using antibodies against RPA2 and RAD51. Representative RPA2/RAD51 foci and DAPI-stained nuclei are shown. **b** Quantification of RPA2/RAD51 foci and nuclear fragmentation. Data represent mean ± SEM of three independent experiments. Over 100 cells were counted in each experiment. **c** Knockdown efficiency of BRCA2 was confirmed by western blotting. **d** BRCA2-deficient VC-8 cells or VC-8 cells reconstituted with wild-type BRCA2 were exposed to 10 Gy IR and then allowed to recover for 2 or 24 h before being processed for immunofluorescence using antibodies against RPA2 and RAD51. Representative RPA2/RAD51 foci and DAPI-stained nuclei are shown. **e** Quantification of RPA2/RAD51 foci and nuclear fragmentation. Data represent mean ± SEM of three independent experiments. Over 100 cells were counted in each experiment. **f** Expression of BRCA2 was confirmed by western blotting. **g** HeLa cells stably expressing GFP-tagged histone H2B (H2B-GFP) were transfected with control siRNA or siRNA against BRCA2. Forty-eight hours post transfection, cells were exposed to 10 Gy IR. Arbitrary cells were selected between 16 and 24 h after IR and performed to visualize chromosomal dynamics by time-lapse imaging (5 min per frame, 0 = 16 h, 51 = 20.25 h). Scale bars, 10 μm

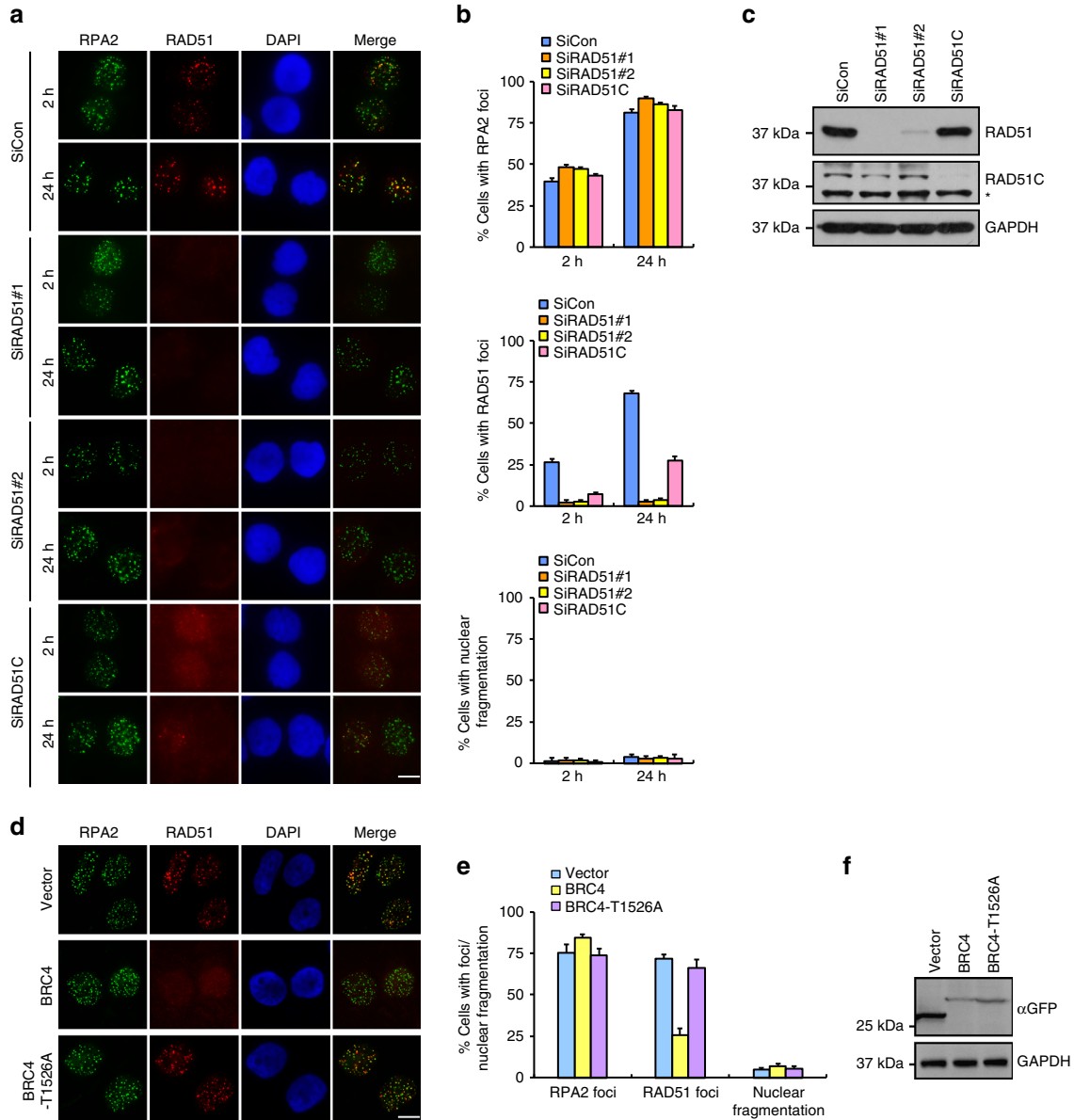

**Fig. 2** BRCA2 suppresses gross genomic instability independently of RAD51. **a** HeLa cells were transfected with control siRNA or siRNAs against RAD51 or RAD51C. Forty-eight hours post transfection, cells were exposed to 10 Gy IR and then allowed to recover for 2 or 24 h before being processed for immunofluorescence using antibodies against RPA2 and RAD51. Representative RPA2/RAD51 foci and DAPI-stained nuclei are shown. **b** Quantification of RPA2/RAD51 foci and nuclear fragmentation. Data represent mean ± SEM of three independent experiments. Over 100 cells were counted in each experiment. **c** Knockdown efficiency of RAD51/RAD51C was confirmed by western blotting. The asterisk indicates a nonspecific band. **d** HeLa cells were electroporated with GFP-tagged wild-type BRC4 or the T1526A mutant. Twenty-four hours later, cells were exposed to 10 Gy IR and then allowed to recover for 24 h before being processed for immunofluorescence using antibodies against RPA2 and RAD51. Representative RPA2/RAD51 foci and DAPI-stained nuclei are shown. **e** Quantification of RPA2/RAD51 foci and nuclear fragmentation. Data represent mean ± SEM of three independent experiments. Over 100 cells were counted in each experiment. **f** BRC4 expression was confirmed by western blotting. Scale bars, 10 μm

Fig. 1d–f, RPA2 foci resolved more readily in VC-8 cells than in VC-8 cells reconstituted with BRCA2.

Given that BRCA2 is involved in the early recruitment of the EXO1 nuclease to DNA damage sites[45], we tested whether loss of EXO1 may recapitulate the BRCA2-associated premature clearance of RPA2 foci. However, in contrast to BRCA2, EXO1 depletion had no appreciable effect on the time-dependent dissolution of RPA2 foci (Supplementary Fig. 1d, e), suggesting that the observed effect of BRCA2 depletion was not due to defective DNA end resection. Our observation that EXO1 depletion alone did not markedly affect DNA end resection and

RPA2 foci formation is consistent with previous reports where DNA2 nuclease can functionally compensate for EXO1[46,47].

**BRCA2 inactivation causes massive nuclear fragmentation.** In addition to accelerating the dissolution of damage-induced RPA2 foci, we found that BRCA2 depletion also caused a significant time-dependent increase in the frequency of nuclear fragmentation following DSB induction (up to 50% at 24 h after IR treatment) (Fig. 1a–c and Supplementary Fig. 1b, c). Notably, treatment of BRCA2-depleted cells with the irreversible pan-caspase inhibitor Z-VAD-FMK was not able to suppress nuclear

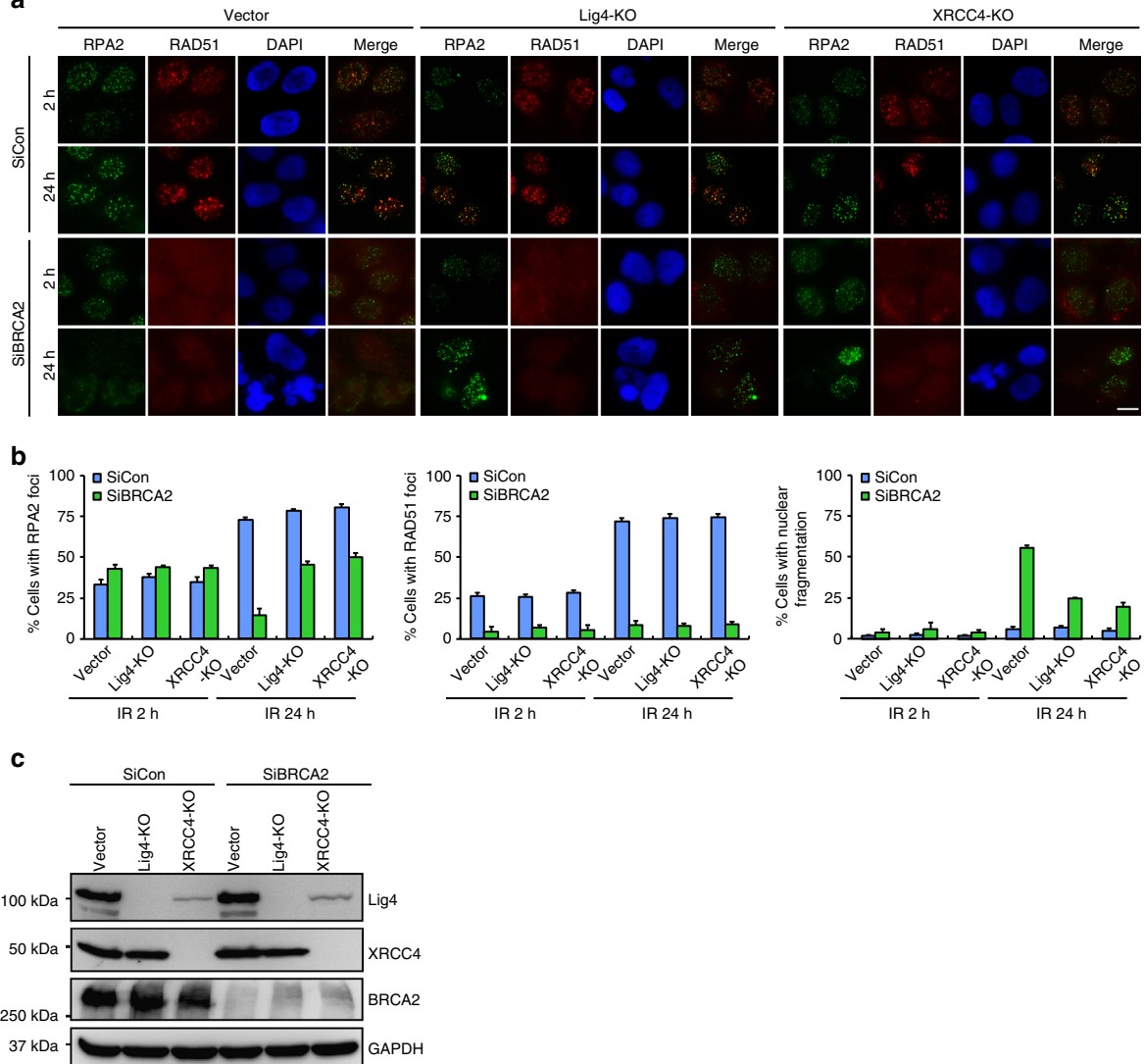

**Fig. 3** C-NHEJ is partially responsible for gross genomic instability in BRCA2-depleted cells. **a** Wild-type, Lig4-, or XRCC4-deficient HeLa cells were transfected with control siRNA or siRNA against BRCA2. Forty-eight hours post transfection, cells were exposed to 10 Gy IR and then allowed to recover for 2 or 24 h before being processed for immunofluorescence using antibodies against RPA2 and RAD51. Representative RPA2/RAD51 foci and DAPI-stained nuclei are shown. **b** Quantification of RPA2/RAD51 foci and nuclear fragmentation. Data represent mean ± SEM of three independent experiments. Over 100 cells were counted in each experiment. **c** Knockdown/knockout efficiency was confirmed by western blotting. Scale bar, 10 μm

fragmentation nor the premature dissolution of RPA2 foci, indicating that these phenotypes likely precede cell apoptosis (Supplementary Fig. 1f, g). The striking time-dependent correlation between the decline in number of RPA2 foci and the increased nuclear fragmentation frequency, and the fact that BRCA2 depletion caused high rates of chromosomal aberrations, including gaps/breaks, radial chromosomes, and translocations, after DSB induction (Supplementary Fig. 1h–j), led us to postulate that, aside its central role in mediating RPA replacement with RAD51, BRCA2 might also prevent inappropriate removal of RPA from ssDNAs to suppress error-prone DSB repair, gross genomic instability, and massive nuclear fragmentation. To test this hypothesis, we performed time-lapse imaging analysis using BRCA2-depleted HeLa cells that stably express H2B-GFP to assess whether nuclear fragmentation observed in BRCA2-depleted cells may result from chromosome missegregation events that occur as a consequence of misrepaired DSBs. Strikingly, we found that the majority of BRCA2-depleted cells

displayed gross abnormalities in chromosome alignment during metaphase (Fig. 1g and Supplementary Movies 1 and 2). Such abnormalities were seldom documented in control cells (Fig. 1g and Supplementary Movies 1 and 2). More importantly, consistent with the notion that BRCA2 contributes to the integrity of the spindle assembly checkpoint[48], BRCA2-depleted cells with misaligned chromosomes entered anaphase prematurely, which in turn resulted in chromosome missegregation and nuclear fragmentation (Fig. 1g and Supplementary Movies 1 and 2).

**BRCA2 but not RAD51 suppresses gross genomic instability.** We next investigated whether RAD51 depletion phenocopies the BRCA2 deficiency-associated change to RPA foci clearance and increase in nuclear fragmentation frequency. Surprisingly, in stark contrast to BRCA2, RAD51 depletion did not noticeably affect the disassembly of RPA2 foci (Fig. 2a–c). In addition, RAD51 depletion did not cause significant increase in the frequency of nuclear fragmentation following IR treatment

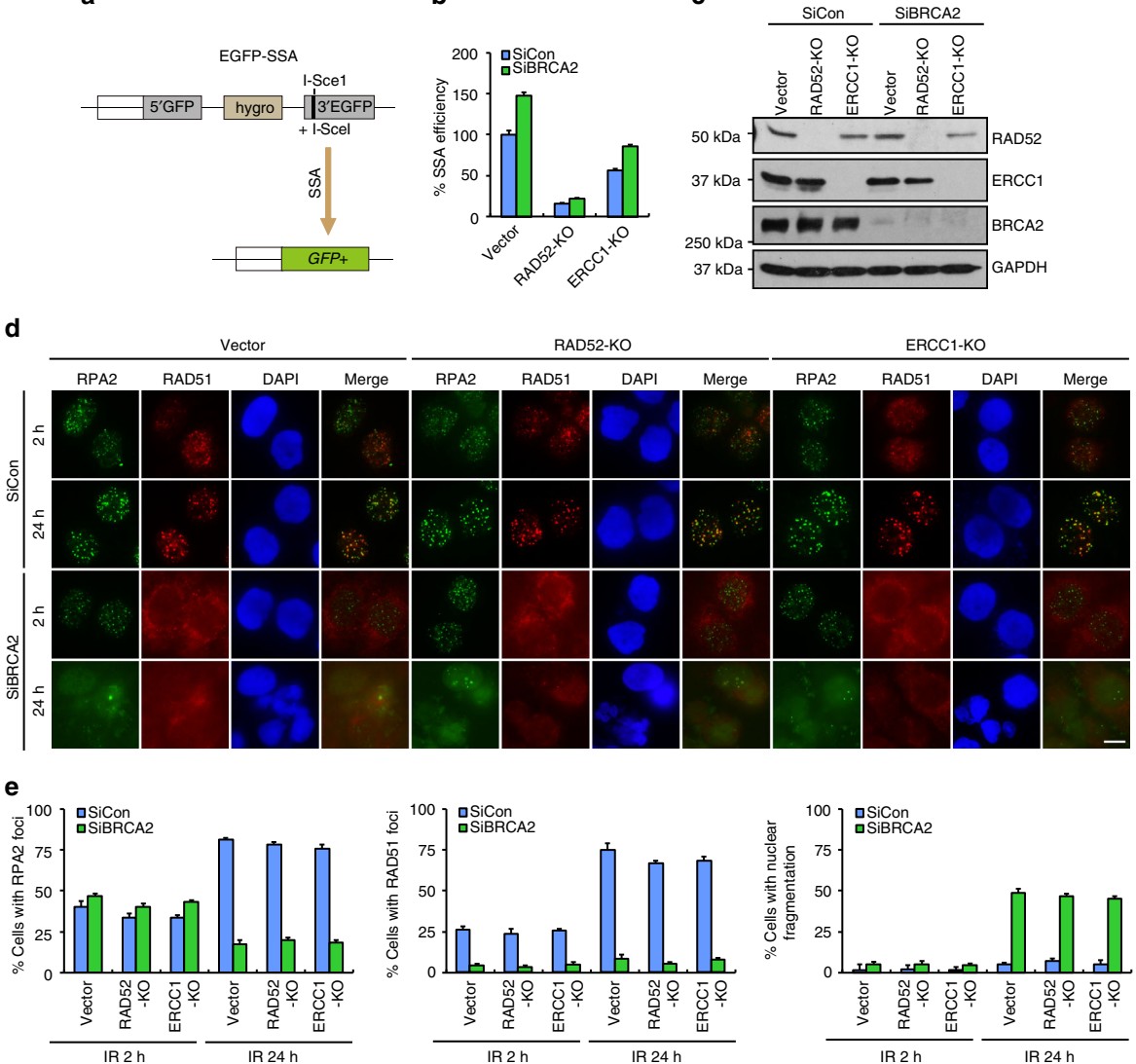

**Fig. 4** SSA is not responsible for gross genomic instability in BRCA2-depleted cells. **a** Schematic representation of the EGFP-based SSA reporter assay. **b** BRCA2 depletion leads to an increased SSA frequency. Wild-type, RAD52-, or ERCC1-deficient U2OS SSA-EGFP cells were transfected with control siRNA or siRNA against BRCA2. Twenty-four hours post transfection, cells were electroporated with pCBASce plasmid. After 48 h, the percentage of GFP-positive cells was determined by FACS. Results represent mean ± SEM of three independent experiments. **c** Knockdown/knockout efficiency was confirmed by western blotting. **d** Loss of RAD52 or ERCC1 was unable to reverse gross genomic instability in BRCA2-depleted cells. Wild-type, RAD52-, or ERCC1-deficient HeLa cells were transfected with control siRNA or siRNA against BRCA2. Forty-eight hours post transfection, cells were exposed to 10 Gy IR and then allowed to recover for 2 or 24 h before being processed for immunofluorescence using antibodies against RPA2 and RAD51. Representative RPA2/RAD51 foci and DAPI-stained nuclei are shown. **e** Quantification of RPA2/RAD51 foci and nuclear fragmentation. Data represent mean ± SEM of three independent experiments. Over 100 cells were counted in each experiment. Scale bar, 10 μm

(Fig. 2a–c). More importantly, whereas overexpression of wild type, but not a mutant allele of the BRCA2 BRC4 repeat significantly impaired RAD51 foci formation as previously reported[49], it had no significant effect on the kinetics of the dissolution of RPA2 foci or on nuclear integrity (Fig. 2d–f). These results suggest that BRCA2 may have a RAD51-independent role in suppressing genomic instability. In line with this hypothesis, overexpression of RAD51 was unable to suppress these defects caused by BRCA2 inactivation (Supplementary Fig. 2a–c).

In addition to BRCA2, the five RAD51 paralogs (RAD51B, RAD51C, RAD51D, XRCC2, and XRCC3) are also required for the assembly of DNA damage-induced RAD51 foci, and mutations in any one of them lead to impaired HR-mediated DSB repair[50]. To further corroborate that BRCA2 suppresses gross genomic instability independently of the RAD51 pathway, we knocked down RAD51C with the RNAi approach. As shown in Fig. 2a–c, whereas RAD51C depletion dramatically reduced RAD51 foci formation as previously established, it had no obvious effect on the kinetics of RPA2 foci nor on nuclear integrity.

**c-NHEJ promotes genome instability in BRCA2-depleted cells.** c-NHEJ has been shown to be the predominant mechanism for joining DSB ends and accounts for chromosomal translocation events in human cells[51]. We thus explored whether c-NHEJ may be responsible for the gross genomic instability observed in BRCA2-deficient cells after DSB induction. To this end, we took

advantage of the CRISPR-Cas9 gene-editing methodology to inactivate the key c-NHEJ components DNA ligase 4 or XRCC4 in HeLa cells. As shown in Fig. 3a–c, unlike BRCA2, inactivation of DNA ligase 4 or XRCC4 alone had no appreciable effect on the kinetics of RPA2 foci nor on nuclear integrity after IR treatment. However, loss of DNA ligase 4 or XRCC4 partially (~60%) suppressed the otherwise premature removal of RPA2 from DNA damage foci and the heightened frequency of nuclear

fragmentation in BRCA2-depleted cells (Fig. 3a–c). More importantly, re-introduction of DNA ligase 4 or XRCC4 into their respective deficient cells reversed the effects of their loss in suppressing these aberrant events observed in BRCA2-depleted cells (Supplementary Fig. 3a–f). Since genetic ablation of key c-NHEJ components did not fully suppress the BRCA2-associated deficits, we propose that, in addition to c-NHEJ, other DSB repair

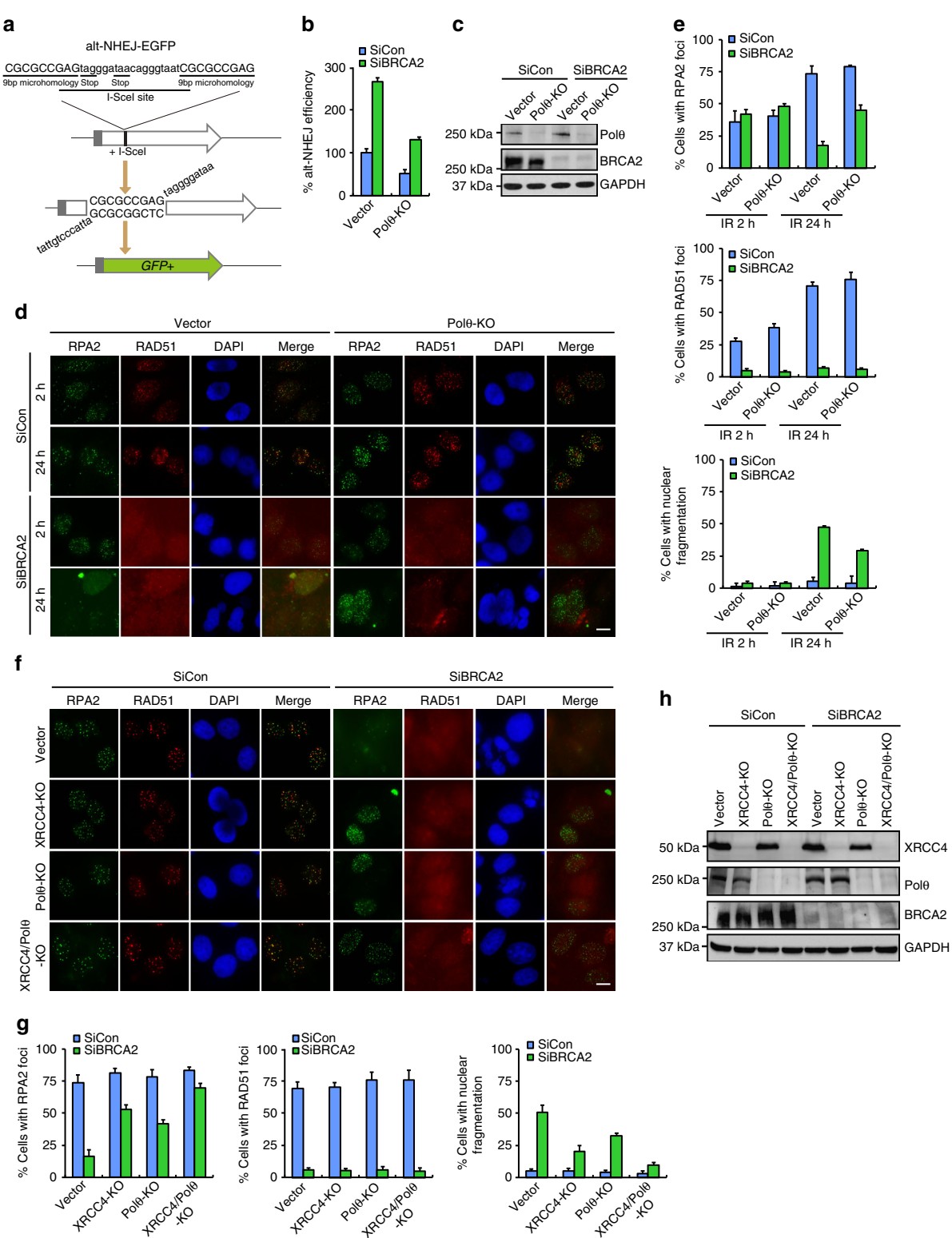

pathway(s) may also contribute to the gross genomic instability observed in BRCA2-depleted cells after DNA damage.

**SSA is dispensable for BRCA2-associated genome instability.** It has been shown that loss of BRCA2 causes misrepair of DSBs bearing repetitive sequence by unleashing the SSA pathway[52,53]. By using a previously established U2OS cell line that harbors a chromosomally integrated copy of the SSA-EGFP reporter to measure SSA-mediated repair of I-SceI-induced DSBs[54], we confirmed that inactivation of BRCA2 indeed triggered a substantial increase in the frequency of SSA (Fig. 4a–c). To determine whether the increased usage of SSA in BRCA2-depleted cells is dependent on RAD52 and ERCC1, two essential components of the SSA pathway, we generated U2OS-SSA-EGFP cells that are nullizygous for either RAD52 or ERCC1 (Fig. 4c). As shown in Fig. 4b, c, SSA frequency in both wild-type and BRCA2-depleted cells was dramatically decreased following the loss of RAD52 and, to a lesser extent, ERCC1.

We next examined whether the gross genomic instability phenotype in BRCA2-depleted cells may result from inappropriate chromosomal DSB repair by SSA. As shown in Fig. 4d, e, neither RAD52 nor ERCC1 silencing was able to stabilize RPA2 foci nor suppress the increased frequency of nuclear fragmentation in BRCA2-depleted cells. These results suggest that, although BRCA2 inactivation resulted in an elevated frequency of SSA, the gross genomic instability observed in BRCA2-depleted cells is not dependent on SSA.

**Alt-NHEJ promotes genome instability in BRCA2-depleted cells.** Like the SSA pathway, the highly error-prone alt-NHEJ pathway has been shown to be aberrantly activated in the absence of BRCA2[55]. Given that c-NHEJ was only partially responsible for the gross genomic instability observed in BRCA2-deficient cells, we considered the possibility that alt-NHEJ may also contribute to the gross genomic instability in cells that lack BRCA2. Using a previously described alt-NHEJ-EGFP reporter system (Fig. 5a)[56], we confirmed that inactivation of BRCA2 indeed resulted in a significant increase in the frequency of alt-NHEJ (Fig. 5b, c). Polθ, an essential component of alt-NHEJ, was used as a positive control to confirm the functionality of this assay (Fig. 5b, c). Notably, the increased alt-NHEJ activity in BRCA2-depleted cells was dependent on Polθ, indicating that BRCA2 is able to suppress Polθ-mediated alt-NHEJ (Fig. 5b, c).

We next examined whether disabling alt-NHEJ may suppress the gross genomic instability caused by BRCA2 inactivation. As shown in Fig. 5d, e, inactivation of the key alt-NHEJ component Polθ partially (~35–40%) suppressed the accelerated dissolution of RPA2 foci and nuclear fragmentation in BRCA2-depleted cells. Consistently, these aberrant events observed in BRCA2-depleted cells were also partially reversed by co-depletion of DNA ligase 1

and DNA ligase 3 (Supplementary Fig. 4a–e). By contrast, inactivation of Polθ or DNA ligases 1/3 did not restore RAD51 foci formation in BRCA2-dpleted cells, further strengthening the idea that the gross genomic instability in BRCA2-depleted cells is not due to defective RAD51 recruitment (Fig. 5d, e and Supplementary Fig. 4a–e). Taken together, these results suggest that toxic alt-NHEJ repair of DSBs also contributes, albeit to a lesser extent than c-NHEJ, to the gross genomic instability observed in BRCA2-deficient cells.

Having established that both c-NHEJ and alt-NHEJ contribute to the premature dissolution of RPA2 foci and nuclear fragmentation in BRCA2-deficient cells, we next sought to determine whether concomitant inactivation of c-NHEJ and alt-NHEJ would fully reverse these aberrant events observed in BRCA2-deficient cells. As shown in Fig. 5f–h, and Supplementary Fig. 4a–e, double knockout of Polθ and XRCC4 or co-inactivation of DNA ligases 1 and 3 and DNA ligase 4 almost completely suppressed the observed defects induced by BRCA2 depletion, indicating that both c-NHEJ and alt-NHEJ pathways contribute to the gross genomic instability observed in BRCA2-deficient cells.

**The nuclease Artemis promotes alt-NHEJ.** Genetic studies in yeast have indicated that the structure-specific endonuclease RAD1-RAD10 (XPF–ERCC1 in vertebrates) participates in alt-NHEJ by catalyzing 3′ flap removal after annealing microhomologies between ssDNA[17,57]. However, loss of ERCC1 had no obvious effect on either basal alt-NHEJ activity or the increased aberrant alt-NHEJ activity associated with BRCA2 deficiency (Fig. 6a). Moreover, unlike Polθ and DNA ligases 1/3, loss of ERCC1 did not reverse the change in the kinetics of the dissolution of RPA2 foci and on nuclear integrity in BRCA2-deficient cells (Fig. 4d, e). Additionally, mouse ES cells deficient in ERCC1 exhibited only a minor alt-NHEJ defect[15]. Together these findings indicated that other nucleases may compensate for the loss of XPF-ERCC1 nuclease activity.

Artemis, originally identified as the gene mutated in one form of human severe combined immunodeficiency (SCID), is responsible for processing DNA ends in c-NHEJ[58]. Since Artemis possesses endonuclease activity capable of cleaving 3′ flaps[59,60], we postulate that it may contribute to the increased aberrant alt-NHEJ activity in BRCA2-depleted cells by cleaving off 3′ flaps from an annealed intermediate. Indeed, loss of Artemis led to a significant reduction in the frequency of alt-NHEJ in BRCA2-depleted cells (Fig. 6b, c). Strikingly, and consistent with results from a previous study[61], loss of Artemis only caused a subtle decrease in the basal levels of alt-NHEJ activity (Fig. 6b, c). These results indicated that Artemis plays a role in alt-NHEJ, especially in the context of BRCA2 deficiency. Importantly, silencing of Artemis and Polθ together did not decrease alt-NHEJ efficiency

**Fig. 5** Alt-NHEJ partially contributes to gross genomic instability in BRCA2-depleted cells. **a** Schematic representation of the EGFP-based alt-NHEJ reporter assay. **b** BRCA2 depletion leads to an increased alt-NHEJ frequency. Wild-type or Polθ-deficient U2OS alt-NHEJ-EGFP cells were transfected with control siRNA or siRNA against BRCA2. 24 h posttransfection, cells were electroporated with pCBASce plasmid. After 48 h, the percentage of GFP-positive cells was determined by FACS. Results represent mean ± SEM of three independent experiments. **c** Knockdown/knockout efficiency was confirmed by western blotting. **d** Loss of Polθ partially suppresses gross genomic instability caused by BRCA2 depletion. Wild-type or Polθ-deficient HeLa cells were transfected with control siRNA or siRNA against BRCA2. Forty-eight hours posttransfection, cells were exposed to 10 Gy IR and then allowed to recover for 2 or 24 h before being processed for immunofluorescence using antibodies against RPA2 and RAD51. Representative RPA2 and RAD51 foci and DAPI-stained nuclei are shown. **e** Quantification of RPA2 and RAD51 foci and nuclear fragmentation. Data represent mean ± SEM of three independent experiments. Over 100 cells were counted in each experiment. **f** Wild-type, Polθ-, XRCC4-, or Polθ/XRCC4-deficient HeLa cells were transfected with control siRNA or siRNA against BRCA2. Forty-eight hours post transfection, cells were exposed to 10 Gy IR and then allowed to recover for 2 or 24 h before being processed for immunofluorescence using antibodies against RPA2 and RAD51. Representative RPA2 and RAD51 foci and DAPI-stained nuclei are shown. **g** Quantification of RPA2 and RAD51 foci and nuclear fragmentation. Data represent mean ± SEM of three independent experiments. Over 100 cells were counted in each experiment. **h** Knockdown/knockout efficiency was confirmed by western blotting. Scale bars, 10 μm

further than was achieved by Artemis or Polθ inactivation alone (Supplementary Fig. 5a, b), suggesting that Artemis and Polθ function in a common pathway to promote alt-NHEJ.

To elucidated whether the nuclease activity of Artemis is critical for its function in promoting alt-NHEJ, we reconstituted Artemis-deficient cells with wild-type Artemis or its nuclease-inactivating mutant H35AD37N and performed rescue

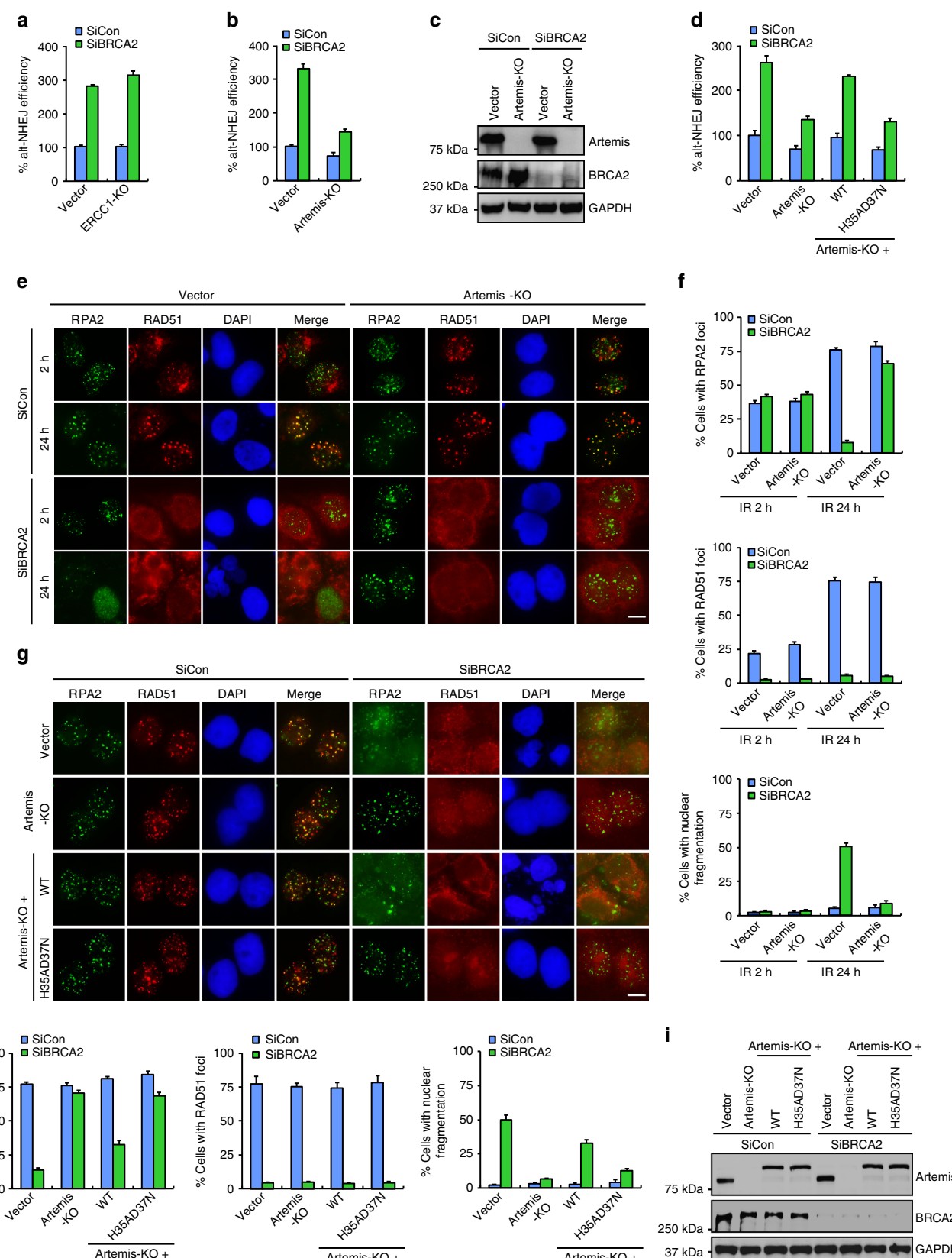

experiments. As shown in Fig. 6d, the effect of Artemis loss in suppressing alt-NHEJ in BRCA2-depleted cells can be reversed by re-introduction of wild-type Artemis but not by the H35AD37N mutant, suggesting that, in addition to its role in c-NHEJ, Artemis also acts as a nuclease to promote DSB repair by alt-NHEJ.

As Artemis plays a critical role in both c-NHEJ and alt-NHEJ repair pathways, we hypothesized that Artemis deficiency may fully suppress the gross genomic instability caused by BRCA2 inactivation. As shown in Fig. 6e, f, loss of Artemis re-stabilized RPA2 at DNA damage foci and suppressed the otherwise elevated incidence of nuclear fragmentation events in BRCA2-deficient cells to similar extents as quantified in cells with concomitant loss of XRCC4 and Polθ. Moreover, re-expression of wild-type Artemis but not its nuclease-inactivated mutant H35AD37N was able to revert the effect of Artemis loss in suppressing these aberrant events observed in BRCA2-inactivated cells (Fig. 6g–i). Collectively, these results suggest that Artemis drives gross genomic instability in BRCA2-deficient cells by promoting both c-NHEJ and c-NHEJ.

**53BP1–RIF1–Artemis axis promotes alt-NHEJ.** A recent study showed that Artemis is a downstream effector of the 53BP1/PTIP-dependent pathway, which trims DNA ends and promotes c-NHEJ[62]. Thus, we decided to determine whether 53BP1 and PTIP, like Artemis, may be responsible for the aberrant activation of the alt-NHEJ pathway in BRCA2-depleted cells. As shown in Fig. 7a, b, downregulation of 53BP1 hampered alt-NHEJ in BRCA2-depleted cells, although to a lesser extent than Artemis silencing. In addition, downregulation of 53BP1 in an Artemis deficient background did not cause further decrease in alt-NHEJ frequency (Fig. 7c). These results suggested that 53BP1 and Artemis stimulated alt-NHEJ in an epistatic manner with resemblance to their interdependent relationship in driving c-NHEJ. More importantly, like Artemis, downregulation of 53BP1 efficiently prolonged the stability of RPA2 foci and restored nuclear integrity in BRCA2-depleted cells (Fig. 7d, e). These observations support the idea that toxic c-NHEJ and alt-NHEJ events may fuel genome instability in BRCA2 null cells, and are responsible for the premature dissolution of RPA2 foci as well as the elevated incidence of nuclear fragmentation following BRCA2 silencing. Surprisingly, however, downregulation of PTIP had no obvious effect on alt-NHEJ in both wild-type and BRCA2-depleted cells, and only marginally reverted the change in the persistence of RPA2 foci and on nuclear integrity in BRCA2 deficient cells (Fig. 7a–e). These results suggested that, although PTIP functions together with 53BP1 to promote the retention of

Artemis at sites of DNA damage, it might play a more specific role in 53BP1/Artemis-dependent c-NHEJ and not in the alt-NHEJ pathway.

Besides PTIP, RIF1 has also been shown to act downstream of 53BP1 to promote c-NHEJ[63–67]. We therefore examined whether RIF1, similar to 53BP1, may play a role in the aberrant activation of the alt-NHEJ pathway in BRCA2-depleted cells. As shown in Fig. 7a, b, downregulation of RIF1 decreased alt-NHEJ frequency in BRCA2-depleted cells to a level similar to that achieved by 53BP1 depletion. Moreover, like 53BP1, depletion of RIF1 in the context of Artemis deficiency did not cause further decrease in alt-NHEJ frequency (Fig. 7c). Importantly, RIF1 depletion efficiently reverted the effects of BRCA2 depletion on the kinetics of RPA2 foci dissolution and on nuclear integrity (Fig. 7d, e). Together, these results suggest that the 53BP1–RIF1 complex promotes DSB repair by alt-NHEJ in BRCA2-depleted cells independently of PTIP.

The aforementioned results prompted us to speculate that RIF1, like 53BP1, may be required for the retention of Artemis at DNA damage sites. To test this hypothesis, we micro-irradiated wild-type or BRCA2-depleted HeLa cells engineered to express GFP-tagged Artemis and monitored its retention at DNA lesions. As shown in Fig. 7f, g, RIF1 depletion hampered the retention of Artemis at laser-generated DSBs. Collectively, these results indicated that the 53BP1–RIF1 complex may promote toxic c-NHEJ and alt-NHEJ events in BRCA2-depleted cells by facilitating the retention of Artemis at sites of DNA damage.

## Discussion

Annealing between exposed homologous DNA molecules is critical for both SSA and alt-NHEJ. In this regard, the RPA heterotrimer binds ssDNA structures at resected DSBs to prevent annealing events that would otherwise lead to SSA or alt-NHEJ[68]. Thus, it is tempting to speculate that overcoming this inhibitory effect of RPA is a prerequisite for ssDNA annealing and hence for DSB repair via the SSA and alt-NHEJ pathways. Consistent with this hypothesis, RAD52 has been shown to efficiently overcome the inhibitory effect of RPA, thereby stimulating ssDNA annealing and DSB repair by SSA[5]. However, RAD52 is dispensable for alt-NHEJ in both *S. cerevisiae* and mammalian cells[13], implying that a hitherto unidentified protein might be specifically involved in facilitating annealing between DNA structures bearing microhomologies. Notably, a recent study has shown that the low fidelity DNA polymerase Polθ is able to promote microhomology annealing and is required for alt-NHEJ[33]. However, since these *in vitro* studies were carried out in

**Fig. 6** Artemis acts as a nuclease to promote alt-NHEJ. **a** ERCC1 is not required for alt-NHEJ. Wild-type or ERCC1-deficient U2OS alt-NHEJ-EGFP cells were transfected with indicated siRNAs and then electroporated with pCBASce plasmid. After 48 h, the percentage of GFP-positive cells was determined by FACS. Results represent mean ± SEM of three independent experiments. **b** Artemis loss reduces the frequency of alt-NHEJ. Wild-type or Artemis-deficient U2OS alt-NHEJ-EGFP cells were transfected with indicated siRNAs and then electroporated with pCBASce plasmid. After 48 h, the percentage of GFP-positive cells was determined by FACS. Results represent mean ± SEM of three independent experiments. **c** Knockdown/knockout efficiency was confirmed by western blotting. **d** The nuclease activity of Artemis is indispensable for its role in promoting alt-NHEJ. Artemis-deficient U2OS alt-NHEJ-EGFP cells stably expressing wild-type Artemis or the nuclease-inactivating mutant (H35AD37N) were transfected with indicated siRNAs and then electroporated with pCBASce plasmid. After 48 h, the percentage of GFP-positive cells was determined by FACS. Results represent mean ± SEM of three independent experiments. **e** Artemis loss suppresses gross genomic instability in BRCA2-depleted cells. Wild-type or Artemis-deficient HeLa cells were transfected with indicated siRNAs, exposed to 10 Gy IR, and then allowed to recover for 2 or 24 h before being processed for immunofluorescence. Representative RPA2/RAD51 foci and DAPI-stained nuclei are shown. **f** Quantification of RPA2/RAD51 foci and nuclear fragmentation. Data represent mean ± SEM of three independent experiments. Over 100 cells were counted in each experiment. **g** The nuclease activity of Artemis is indispensable for its role in promoting gross genomic instability in BRCA2-deficient cells. Artemis-deficient HeLa cells stably expressing wild-type Artemis or the nuclease-inactivating mutant (H35AD37N) were transfected with indicated siRNAs, exposed to 10 Gy IR, and then allowed to recover for 24 h before being processed for immunofluorescence. Representative RPA2/RAD51 foci and DAPI-stained nuclei are shown. **h** Quantification of RPA2/RAD51 foci and nuclear fragmentation. Data represent mean ± SEM of three independent experiments. Over 100 cells were counted in each experiment. **i** The expression of wild-type Artemis or its H35AD37N mutant was confirmed by western blotting. Scale bars, 10 μm

the absence of RPA[33], it remains unclear whether the inhibitory effect of RPA on microhomology annealing is indeed overcome by Polθ. We posit that certain enzymatic activity should displace RPA from ssDNA structures prior to PARP1/Polθ-mediated microhomology annealing.

At resected DSBs, BRCA2 facilitates the RPA to RAD51 exchange and promotes formation of RAD51-ssDNA filaments. RAD51 nucleofilaments then catalyze strand invasion into homologous duplex DNA, leading to the formation of transient displacement loops (D-loops). It is noteworthy that this event is

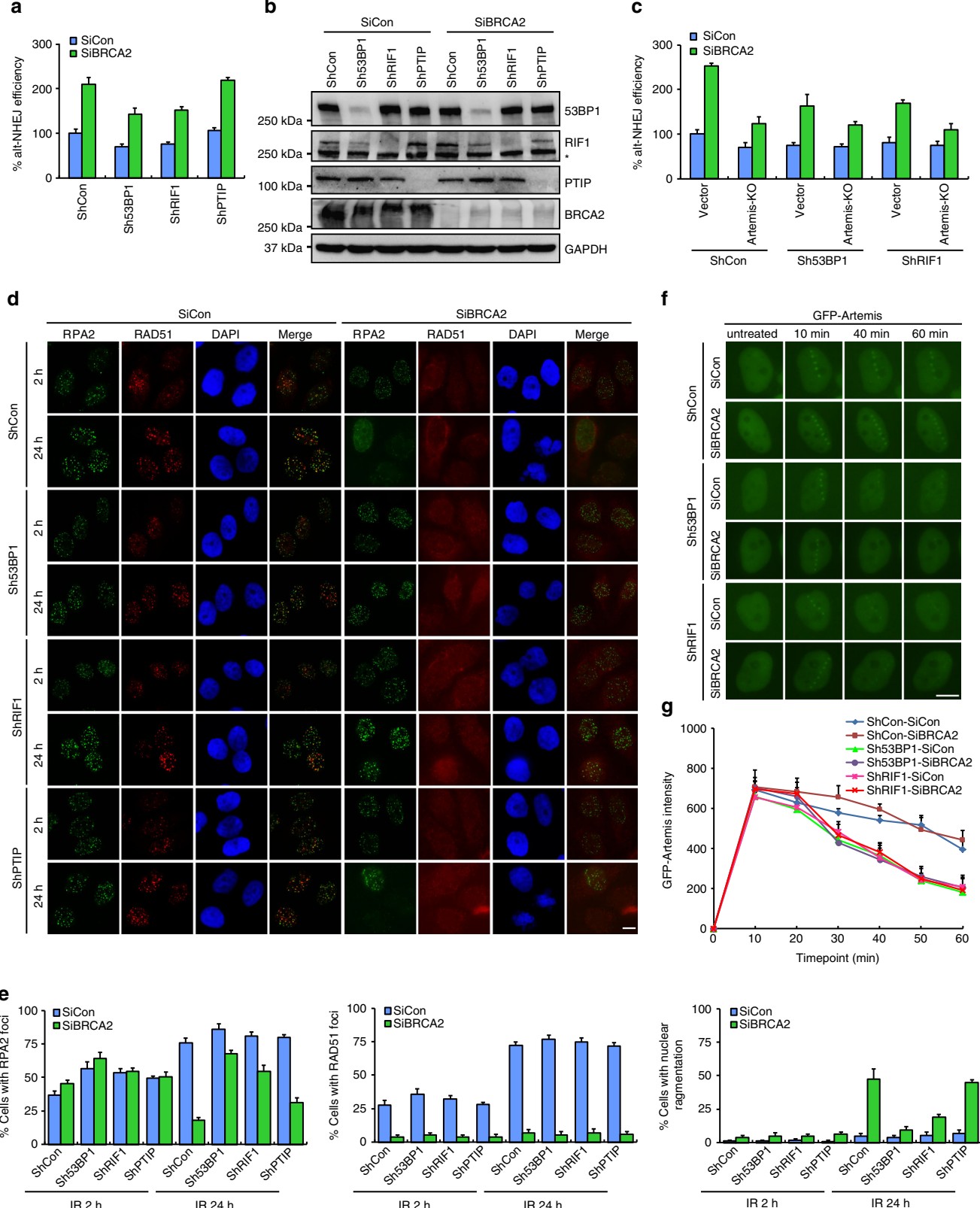

accompanied by RPA binding to the displaced strand within the D-loop, resulting in the stabilization of the invasion intermediate. Thus, one would envisage that RPA foci may decline only when D-loops are resolved upon completion of an HR reaction. In sharp contrast to BRCA2-assisted displacement of RPA by RAD51 during HR repair, both RAD52- and Polθ/unknown protein-mediated RPA release from ssDNAs during SSA and alt-NHEJ, respectively, may result in homology annealing and 3′ flap cleavage, and therefore may correspond to the more immediate RPA foci decline. In this study, we found that inactivation of BRCA2 but not RAD51 led to the premature dissolution of DNA damage-induced RPA2 foci. Taking into account that BRCA2 was rapidly recruited to sites of DNA damage (Supplementary Fig. 6) and its deficiency resulted in substantially increased frequency of alt-NHEJ and SSA repair events, we proposed that BRCA2 may, in addition to its direct role in facilitating RAD51 nucleoprotein filament assembly, also restrict alt-NHEJ and SSA repair by antagonizing Polθ/unknown protein- and RAD52-mediated RPA displacement from ssDNAs. Surprisingly, overexpression of RPA was unable to stabilize RPA foci or suppress the increased frequency of nuclear fragmentation in BRCA2-depleted cells (Supplementary Fig. 7a–c), indicating that, even in the presence of excess RPA, Polθ/unknown protein and RAD52 are still able to release RPA from ssDNA in the absence of BRCA2. Intriguingly, although inactivation of BRCA2 resulted in an elevated frequency of SSA, loss of key components in the SSA pathway did not restore the effects of BRCA2 deficiency on the kinetics of RPA2 foci dissolution and on nuclear integrity. These findings may reflect that only a very small fraction of IR-induced DSBs in BRCA2-deficient cells is repaired by SSA.

After microhomology annealing, the remaining non-complementary 3′ flaps must be removed to produce ends suitable for gap-filling and ligation. In yeast, this function is fulfilled by the structure-specific RAD1–RAD10 endonuclease[17,57]. Strikingly, inactivation of the RAD10 mammalian homolog ERCC1 in mouse embryonic stem cells[15] or in human U2OS cells (Fig. 6a) did not noticeably affect the frequency of alt-NHEJ. Moreover, unlike the key alt-NHEJ factors Polθ and DNA ligases 1/3, loss of ERCC1 was unable to reverse the effects of BRCA2 depletion on the kinetics of RPA2 foci dissolution and on nuclear integrity. These findings support the existence of a hitherto unknown nuclease that can catalyze 3′ flap removal after annealing microhomologies between ssDNA molecules in BRCA2-deficient cells. In this study, we found that loss of Artemis efficiently reverted the effect of BRCA2 depletion on alt-NHEJ. In addition, the ability of Artemis to promote alt-NHEJ repair in BRCA2-depleted cells is strictly dependent on its nuclease activity. These results, taken together with the

observation that Artemis is endowed with activity to cleave DNA flap structures[59,60], indicate that Artemis may drive aberrant activation of alt-NHEJ in BRCA2-depleted cells by removing 3′ flaps from alt-NHEJ intermediates. Strikingly, similar to alt-NHEJ, a recent study showed that resection-dependent c-NHEJ in G1 cells may also use microhomologies to align the DSB ends prior to ligation and requires the Artemis nuclease[69].

In contrast to HR, alt-NHEJ has been shown to occur in the context of a shorter ssDNA overhang and requires only limited DNA end resection[57]. Interestingly, a recent study found that alt-NHEJ can also mediate repair across longer distance[68]. As BRCA2 inactivation had no significant effect on DNA end processing, we propose that the extensive resection in BRCA2-deficient cells may enhance the chance to expose microhomologies that are distant from DSB sites, thereby promoting the use of distal microhomologies for alt-NHEJ repair. Given that the long non-complementary 3′ ssDNA tails adjacent to the distal microhomologies may destabilize annealing between microhomologies and reduce alt-NHEJ frequency, it is possible that a large portion of the long non-complementary 3′ ssDNA are removed before microhomology annealing. As Artemis can cleave both overhangs and flaps[59,60], we speculate that it may have a dual role in promoting alt-NHEJ in BRCA2-deficient cells: (1) by cleaving a large portion of the long non-complementary 3′ ssDNA overhangs before microhomology annealing; and (2) by removing short 3′ flaps after microhomology annealing. However, we cannot rule out the possibility that this orchestrated process is accomplished through the concerted action of Artemis and other DNA nucleases. This intriguing possibility requires further experimental investigation.

It is well established that the Artemis nuclease trims DNA ends and promotes c-NHEJ. In this study, we found that Artemis contributes to toxic alt-NHEJ events in BRCA2-depleted cells. Artemis promotes alt-NHEJ in manners that required its nuclease activity, and was dependent on 53BP1 and its downstream effector RIF1 but independent of PTIP. The observation that loss of Artemis, 53BP1, or RIF1 faithfully suppressed the gross genomic instability caused by BRCA2 inactivation raises the possibility that 53BP1-RIF1-Artemis axis may drive gross genomic instability in BRCA2-deficient cells via two distinct mechanisms: (1) by promoting c-NHEJ, and (2) by promoting alt-NHEJ. In complete agreement with this prediction, while inhibition of either c-NHEJ or alt-NHEJ alone partially reverted the gross genomic instability caused by BRCA2 depletion, concomitant inactivation of c-NHEJ and alt-NHEJ almost fully suppressed these aberrant events observed in BRCA2-depleted cells. Based on these findings, we proposed that, if the long ssDNA overhangs generated by extensive resection in BRCA2-

**Fig. 7** Artemis functions epistatically with 53BP1 and RIF1 to promote alt-NHEJ. **a** 53BP1 or RIF1 depletion reduces the frequency of alt-NHEJ in both wild-type and BRCA2-depleted cells. 53BP1-, RIF1- or PTIP-depleted U2OS alt-NHEJ-EGFP cells were transfected with indicated siRNAs and then electroporated with pCBASce plasmid. After 48 h, the percentage of GFP-positive cells was determined by FACS. Results represent mean ± SEM of three independent experiments. **b** Knockdown efficiency was confirmed by Western blotting. The Asterisk indicates a nonspecific band. **c** Artemis functions epistatically with 53BP1 and RIF1 to promote alt-NHEJ. Wild-type or Artemis-deficient U2OS alt-NHEJ-EGFP cells infected with the indicated lentiviral shRNAs were transfected with control siRNA or siRNA against BRCA2. Twenty-four hours post transfection, cells were electroporated with pCBASce plasmid. After 48 h, the percentage of GFP-positive cells was determined by FACS. Results represent mean ± SEM of three independent experiments. **d** Loss of 53BP1 or RIF1 suppresses gross genomic instability in BRCA2-depleted cells. Wild-type, 53BP1-, PTIP-, or RIF1-deficient HeLa cells were transfected with indicated siRNAs, exposed to 10 Gy IR, and then allowed to recover for 2 or 24 h before being processed for immunofluorescence using antibodies against RPA2 and RAD51. Representative RPA2 and RAD51 foci and DAPI-stained nuclei are shown. **e** Quantification of RPA2/RAD51 foci and nuclear fragmentation. Data represent mean ± SEM of three independent experiments. Over 100 cells were counted in each experiment. **f** RIF1 is required for the retention of Artemis at laser-induced DSBs. 53BP1- or RIF1-depleted HeLa cells were transfected with indicated siRNAs. Forty-eight hours later, cells were transfected with GFP-Artemis and were then micro-irradiated and monitored by live cell imaging. Representative images taken at the indicated times after laser microirradiation are shown. **g** The fluorescence intensity at the microirradiated site was quantified. Data represent mean ± SEM from at least 20 cells in three independent experiments. Scale bars, 10 μm

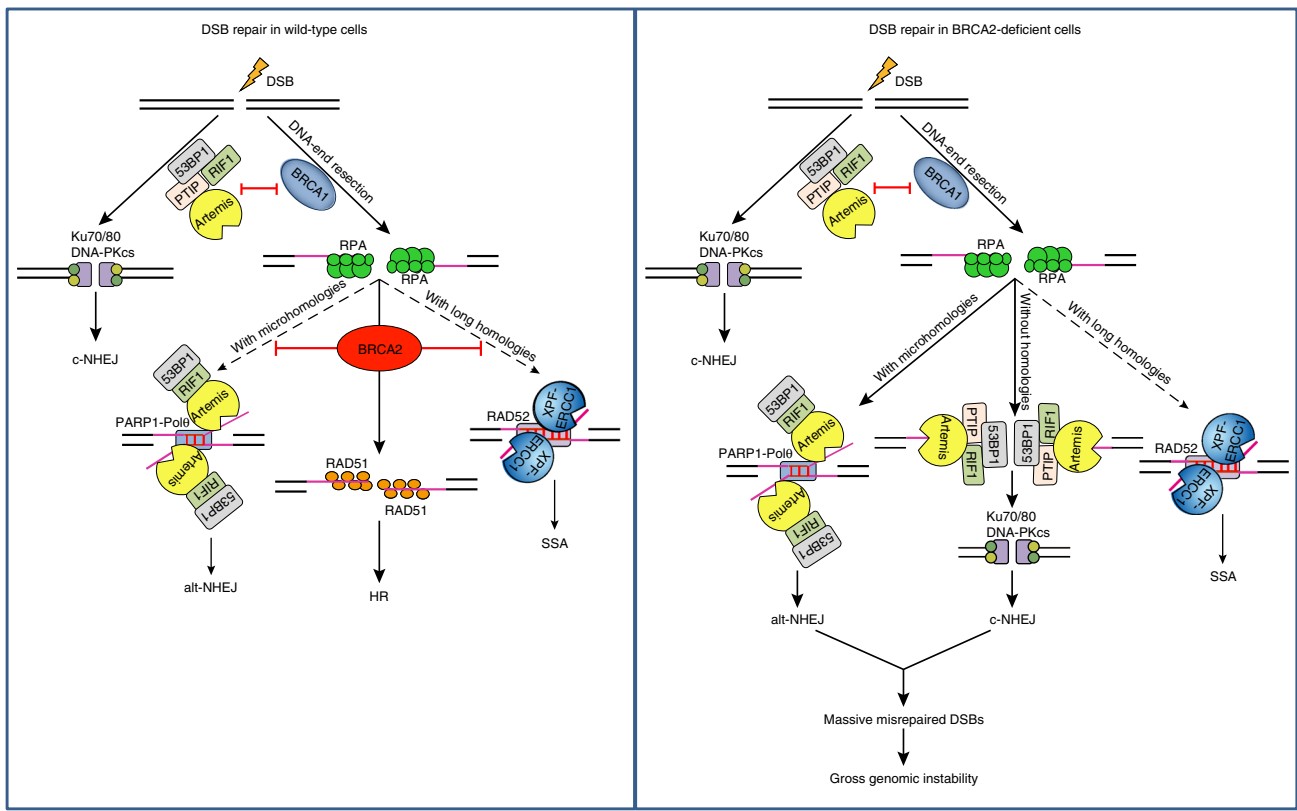

**Fig. 8** A model showing the role of BRCA2 in maintaining genome stability

deficient cells contain short regions of homology, they can anneal to each other at microhomologous regions. After that, the remaining non-complementary 3′ flaps are removed by Artemis, and alt-NHEJ is completed by gap filling DNA synthesis and ligation (Fig. 8). However, if resection fails to expose any DNA stretches with microhomologies, Artemis would process the long 3′ ssDNA overhangs to generate substrates suitable for c-NHEJ (Fig. 8). Thus, DSBs in BRCA2-depleted HR-compromised cells are diverted to repair via the c-NHEJ and alt-NHEJ pathways. Since c-NHEJ- and alt-NHEJ-mediated rejoining of DSBs, especially one-ended DSBs generated by replication fork collapse/stalling, can lead to deletions at DSB sites and/or genetic rearrangements, we posit that the toxic c-NHEJ and alt-NHEJ events contribute to gross genomic instability in BRCA2-depleted cells (Fig. 8).

Notably, while PTIP is not involved in 53BP1/RIF1/Artemis-mediated processing of long ssDNA generated by extensive resection in BRCA2-defecient cells, it contributes to the degradation of nascent DNA in BRCA2-deficient cells by recruiting the Mre11 nuclease to stalled replication forks[40]. Moreover, the function of PTIP at stalled replication forks appears to be independent of its interactions with 53BP1 and RIF1[40]. Based on these findings, we speculate that BRCA2 has at least two HR-independent functions: (1) it protects nascent DNA strands from PTIP/Mre11-mediated degradation at stalled replication forks; (2) it protects the long ssDNA generated by extensive resection from 53BP1/RIF1/Artemis-mediated degradation at DSB sites.

## Methods

**Antibodies**. Rabbit polyclonal anti-BRCA2 (1:200), anti-RAD51 (1:1,000), anti-RAD52 (1:500), anti-RIF1 (1:200), and anti-Polθ (1:200) antibodies were generated by immunizing rabbits with GST-BRCA2 (residues 2800–3000), MBP-RAD51,

GST-RAD52, MBP-RIF1 (residues 1367–1461), and MBP-Polθ (residues 2125–2320) fusion proteins expressed and purified from *E. Coli*, respectively. Anti-ERCC1 (1:2000, ab129267), anti-RPA2 (1:1000, ab2175), anti-Lig1 (1:2000, ab177946), anti-Lig4 (1:2000, ab193353) and anti-53BP1 (1:20,000, ab175188) antibodies were purchased from Abcam. Anti-GAPDH (1:3000, MAB374) and anti-RAD51C (1:1000, NB100–177) antibodies were purchased from Millipore and Novus, respectively. Anti-Lig3 (1:2000, 611876) and anti-XRCC4 (1:500, sc-8285) antibodies were purchased from BD biosciences and Santa Cruz, respectively. Anti-Artemis (1:3000) and anti-PTIP (1:2000) antibodies were gifts from Dr. Junjie Chen and Dr. Zihua Gong (MD Anderson Cancer Center). Uncropped immunoblots are shown in Supplementary Fig. 8.

**Cell culture**. VC-8 and VC-8 + BRCA2 CHO cell lines were a gift from Dr. Bing Xia at Rutgers Cancer Institute of New Jersey. U2OS SSA-EGFP and U2OS alt-NHEJ-EGFP cell lines were gifts from Dr. Xiaohua Wu at Scripps Research Institute. HeLa H2B-GFP cell line was a gift from Dr. Fangwei Wang at Zhejiang University. HEK293T, HeLa, and U2OS cell lines were purchased from ATCC. These cell lines were cultured in Dulbecco's modified essential medium (DMEM) supplemented with 10% fetal bovine serum (FBS) and 1% penicillin and streptomycin. All cell lines used in this study have been cultured in the presence of antimycoplasma antibiotics (BioMycoX Mycoplasma Elimination kit, BioInd) to prevent mycoplasma contamination.

**Plasmids and transfection**. The BRCA2 BRC4 (1511aa–1575aa) with N-terminal NLS sequence KRKRKR were amplified by PCR and cloned into the GFP-C1 vector. All other cDNAs were amplified by PCR and subcloned into the pDONR201 vector according to the manufacturer's instructions (Invitrogen). The resulting Gateway entry clones were recombined into Gateway destination vectors via LR reactions. Transient plasmid transfection was carried out using Lipofectamine 2000 (Invitrogen) according to the manufacturer's guidelines.

**CRISPR-Cas9 gene editing**. All knockout cell lines (HeLa, U2OS SSA-EGFP, or U2OS alt-NHEJ-EGFP) used in this study were generated using CRISPR-Cas9 gene editing approach. The guide RNA (gRNA) sequences were as follows: Lig4-guide A: TCTGAGTTATAAGTTGAAGA, guide B: GCTTATACGGATGATCATAA; XRCC4-guide: TACTGGGTTCAGAAACAAGG; RAD52-guide A: CTGCAGC TGTGGGTGTC CCA, guide B: TGCTGTGATACCGGCG GACC; ERCC1-guide A: TTCCTTGCTGGCGGCCCTGA, guide B: ATTTGTGATACC CCTCGACG; Artemis-guide A: AACAACAACTCCTTAGTCAC, guide B: CGAGCCC GAAAT

ACAGATTT; Polθ-guide A: GTAGA GTTCAGCATTCAACC, guide B: CTTT CGCCCTGTACCGCTTTT. The gRNA sequences were cloned into the plasmids pSpCas9n(BB)-2A-Puro (PX462) or PX330-U6-Chimeric_BB-CBh-hSpCas9 (gifts from Dr. Feng Zhang). The gRNA/Cas9 expression constructs were transient transfected into HeLa, U2OS SSA-EGFP, or U2OS alt-NHEJ-EGFP cells. Twenty-four hours after transfection, cells were cultured in the presence of puromycin (2 μg ml⁻¹) for 48 h prior to plating for individual clones. Knockout cells were validated by western blotting and sequencing.

**siRNAs and shRNAs**. The siRNA target sequences were as follows: Con, UUCAAUAAAUUCUUGAGGUUU; BRCA2#1, GAAGAAUGCAGGUUUAA UAdTdT; BRCA2#2, GGGAAACACUCAGA TTAAAdTdT; EXO1, CAAGCCUA UUCUCGUAUUUdTdT; RAD51#1, CUGUCUA CUGGAACAAUCUUdTdT; RAD51#2, CCUAUUGGAGGAAAUAUCAdTdT; RAD51C: GUACAGCACUGG AACUUCUUU; Lig1: AAGGGCAAGACAGCAGAGGCCdTdT; Lig3: CCACAA AAAAAATCGAGGAdTdT. siRNA transfection was performed using Lipofecta-mine RNAiMAX transfection reagent (Invitrogen) according to the manufacturer's instructions. The shRNA target sequences were as follows: Control shRNA 5′-CC CATAAGAGTAATAATAT-3′; 53BP1 shRNA 5′-GATACTCCTTGCCTGATA A TT-3′; RIF1 shRNA 5′-CGCATTCTGCTGTTGTTGATT-3′; PTIP shRNA 5′-G TGTTTG CAATTGCGGATTAT-3′. Lentiviral shRNA constructs were co-transfected with the packaging plasmids pMD2G and pSPAX2 (gifts from Dr. Songyang Zhou at Baylor College of Medicine) into HEK293T cells. Forty-eight hours after transfection, the resulting lentiviral particles were used to transduce HeLa or U2OS alt-NHEJ-EGFP cells in the presence of 8 μg ml⁻¹ polybrene (Sigma). Cells containing stably integrated shRNA expression constructs were selected in medium containing 2 μg ml⁻¹ puromycin (Calbiochem).

**Immunofluorescent staining**. HeLa cells cultured on coverslips were treated with 10 Gy IR or 0.5 μM CPT for the indicated times. Cells were then washed with PBS, pre-extracted with 0.5% Triton X-100 solution for 5 min, and fixed with 3% par-aformaldehyde for 10 min at room temperature. After blocking with 1% BSA, cells were incubated with indicated primary antibodies for 20 min at room temperature. Subsequently, cells were washed three times with PBS and incubated with indicated secondary antibodies for 20 min at room temperature. After that, cells were stained with DAPI and visualized by a fluorescence microscope (Eclipse 80i; Nikon) equipped with a Plan Fluor ×60 oil objective lens (NA 0.5–1.25; Nikon) and a camera (CoolSNAP HQ[2]; PHOTOMETRICS).

**SSA and alt-NHEJ reporter assays**. U2OS cells with SSA-EGFP or alt-NHEJ-EGFP integration were gifts from Dr. Xiaohua Wu at Scripps Research Institute. 1 × 10⁶ U2OS SSA-EGFP or alt-NHEJ-EGFP cells were electroporated with 12 μg I-SceI expression plasmid (pCBASce). Forty-eight hours post pCBASce electro-poration, cells were harvested, and the expression of GFP was analyzed by flow cytometry.

**Lentivirus packaging and infection**. Artemis entry clone was transferred into a lentiviral Gateway destination vector pCMV-SFB. Lentiviral particles were pro-duced in HEK293T cells by transient co-transfection of the SFB-Artemis lentiviral expression plasmid with packaging plasmids pMD2G and pSPAX2 (gifts from Dr. Songyang Zhou at Baylor College of Medicine). Forty-eight hours after transfec-tion, the resulting lentiviral particles were used to transduce HeLa or U2OS alt-NHEJ-EGFP cells in the presence of 8 μg ml⁻¹ polybrene (Sigma). Cells stably expressing SFB-Artemis were established by selection in medium containing 2 μg ml⁻¹ puromycin (Calbiochem).

**Retrovirus production and infection**. Lig4 or XRCC4 entry clone was transferred into a retroviral Gateway destination vector pEF1A-HA-Flag. Retroviruses were produced by transient co-transfection of HEK293T cells with the retroviral expression plasmid, the pCL-ECO packaging plasmid, and the VSV-G envelop expressing plasmid. Viral supernatants were collected at 48 h post transfection and were used to transduce Lig4-deficient or XRCC4-deficient HeLa cells in the pre-sence of 8 μg ml⁻¹ polybrene (Sigma). Cells stably expressing HA-Flag-Lig4 or HA-Flag-XRCC4 were established by selection in medium containing 2 μg ml⁻¹ pur-omycin (Calbiochem).

**Detection of chromosome aberrations and M-FISH**. Forty-eight hours after the siRNA transfection, HeLa cells were exposed to IR (10 Gy) and allowed to recover for 18 h. Cells were then treated with 0.4 μg ml⁻¹ Colcemid (Sigma) for 2 h and swollen using 75 mM KCl for 15 min at room temperature. After fixing with 3:1 (vol/vol) methanol/acetic acid for 20 min, cells were dropped onto wet slides at room temperature and air-dried. Subsequently, slides were stained with 5% Giemsa stain solution for 5 min and examined under a light microscopy. Fifty metaphases per sample were scored for the number of chromosome aberrations. M-FISH was performed following the manufacturer's instructions (MetaSystems).

**Laser micro-irradiation and live-cell imaging**. HeLa cells expressing GFP-Artemis were seeded onto 35-mm glass-bottomed dishes. Twenty-four hours later,

cells were micro-irradiated using an inverted fluorescence microscope (Eclipse Ti-E; Nikon) equipped with a computer-controlled MicroPoint laser Ablation System (Photonics Instruments; 365 nm, 20 Hz). Time-lapse live-cell images were captured using the same microscope controlled by MetaMorph imaging software (Molecular Devices).

**Time-lapse live-cell imaging**. HeLa cells stably expressing H2B-GFP (a gift from Dr. Fangwei Wang at Zhejiang University) were transfected with indicated siRNAs and then plated in 35-mm 4-chamber glass-bottomed dishes. Twenty-four hours later, cells were exposed to 10 Gy IR and allowed to recover for 18 h. Time-lapse live-cell images were taken with the GE DV Elite Applied Precision Delta Vision system (GE Healthcare) equipped with Olympus oil objectives of ×40 (NA 1.42) Plan Apo N and an API Custom Scientific CMOS camera, and Resolve3D soft-WoRx imaging software. Cells were filmed in a climate-controlled and humidified environment (5% CO₂ at 37 °C). Images were captured every 5 min for 6 h (from 18 to 24 h after IR treatment). The acquired images were processed using Adobe Photoshop CS5 software.

**Statistics and reproducibility**. All experiments were repeated at least twice and similar results were observed. Information about statistical tests and how many times individual experiments were repeated is provided in the Figure legends or respective Methods sections.

**Data availability**. All relevant data supporting the findings of this study are available upon request from the corresponding author.

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

## Acknowledgements

We thank Dr. Xiaohua Wu for SSA-EGFP and alt-NHEJ-EGFP integrated U2OS cell lines, Dr. Bing Xia for VC-8 CHO cell lines, Dr. Fangwei Wang for HeLa cells stably expressing H2B-GFP, and Dr. Junjie Chen and Dr. Zihua Gong for anti-PTIP and anti-Artemis antibodies. This work was supported by the Key Program of the National Natural Science Foundation of China (31730021), National Natural Science Funds for Distinguished Young Scholar, National Program for Special Support of Eminent Professionals, and National Natural Science Foundation of China Grant 81661128008 and 31571397. J.J.H. was supported by National Postdoctoral Program for Innovative Talents (BX201600134).

## Author contributions

J.J.H., T.L., M.S.Y.H. and J.H. designed the experiments; J.J.H. and C.R. performed the experiments; J.W., A.X. and C.F. provided essential experimental materials; J.J.H., C.R., T.L. and J.H. analyzed the data; J.H. and T.L. wrote the manuscript.

## Additional information

**Competing interests:** The authors declare no competing financial interests.

