## [Peer Review File · Nature Communications]

Reviewers' comments:

Reviewer #1 (Remarks to the Author):

It has also previously been shown that BRCA2 has a RAD51-independent role in blocking stalled replication fork degradation by MRE11 (see comment below). This latter role is less well characterised and is the focus of this paper. Here it is shown that in the absence of BRCA2 the broken DNA is rejoined by MMEJ and c-NHEJ (and not by RAD52-dependent SSA as might be expected) with a requirement for Artemis and promoted by 53BP1/RIF1. These represent important additions to our understanding of how mis-rejoining occurs when DNA is nucleolytically degraded at the fork and generally I find the work interesting and novel. I have two major comments – firstly the paper does not discuss the known literature adequately. Secondly, the nature of MMEJ is not sufficiently clear and an additional experiment could add clarity. These are given in points below.

1. There have been several papers reporting the role of BRCA2 (distinct from HR) in fork protection from MRE11. The first paper was by Schlacher and Jasin (PMID:21565612). There are several additional papers since then. eg PMC4973925 by Nuzzensweig (which reports a role for PTIP in this process.) These papers must be cited. This does not negate the work here, which rather describes the downstream consequences of fork degradation. Figures 1 are needed but how they overlap with known findings should be stated. This aspect is not novel and should not be presented as such. Thus, the entire first part of the paper should be rewritten. Additionally, the paper by Nussensweig describes a role for PTIP in this process, which is distinct to what is found here. These differences should be discussed. (PMC4614542 FANCD2, FANCI and BRCA2 cooperate to promote replication fork recovery independent of the FA core complex). The novelty of this work should be clearly stated – ie defining how the degraded DNA is rejoined.

2. In the literature MMEJ is often taken to represent Alt-NHEJ (as stated here). However, a recent study by the Lobrich laboratory (Biehs et al; PMID: 28132842) and others have shown that c-NHEJ can also rejoin using small microhomology (MH, which I believe is a now recognized concept). Generally, Alt-NHEJ uses longer regions of MH (still shorter than HR) whereas c-NHEJ uses shorter MH but the precise overlap is unclear - whether c-NHEJ can use 9bp (as occurs in the reporter assay) is unclear. As discussed in the Biehs et al paper, MMEJ represents rejoining involving MH – it can arise via Alt-NHEJ, c-NHEJ or RAD52-dependent SSA. Thus, distinguishing MMEJ and c-NHEJ is misleading – the more meaningful distinction is Alt-NHEJ and c-NHEJ. The requirement for Pol theta and the additivity of pol theta/XRCC4 loss is the strongest evidence that the process reflects Alt-NHEJ (Artemis appears to have functions in both c-NHEJ and Alt-NHEJ). However, whether pol theta actually functions in some c-NHEJ has not been well examined, since this process is poorly understood. The most informative analysis would be to examine the ligase required since this distinguishes the processes. Alt-NHEJ is ligase I/III dependent whereas c-NHEJ is ligase IV dependent. A partial requirement for ligase IV is already shown. Thus, it would be beneficial to examine siRNA of ligase I/III and/or PARP inhibition. I do not feel KO of ligase I/III is required (ie siRNA should suffice) (nor dissection of the individual contribution of ligase I/III) but merely the demonstration that these ligases contribute especially when ligase IV is absent (and that there is complete loss when both are absent). The analysis of RAD52 argues that the rejoining process is not SSA. This would then allow the conclusion that the rejoining processes are Alt-NHEJ and c-NHEJ with MH usage and Artemis-dependency. The MH usage could be discussed briefly. This avoids the conclusion (as currently presented) that c-NHEJ does not involve MH – which is not known. Biehs et al. It would be useful to have this put in context in the discussion.

3. Extending from this, I am somewhat confused by the finding the nuclear fragmentation is abolished when LigIV/XRCC4 are absent. If c-NHEJ has a function, and if RPA foci remain, surely there should be more nuclear fragmentation – maybe not aberrantly rejoined fragments (ie rearrangements) but certainly unrejoined fragments. This is confusing and should be discussed – can unrejoined fragments be distinguished from rearrangements? The figure presented appears to show many unrejoined fragments - so why are they lost when XRCC4/LigIV is absent?

4. The role for 53BP1 here is surprising and distinct to other roles of 53BP1, which is to protect

against resection – here actually loss of 53BP1 prevents degradation (MRE11-dependent) ie RPA foci loss. Indeed, 53BP1 is often described as being pro-NHEJ whereas in this case it appear to protect against NHEJ usage. This could be usefully brought out in the discussion.

Reviewer #2 (Remarks to the Author):

This paper describes the consequence of BRCA2 loss. Surprisingly this leads to a reduction in RPA foci, the exploration in the manuscript as to the pathways involved reveals NHEJ and MMEJ as those responsible. (As might be expected SSA is not). Moreover BRCA2 depleted cells exhibit severe chromosome fragmentation that is also a result of MMEJ and NHEJ usage. They reveal that ARTEMIS in particular plays a role.

The most surprising insight for me comes in the discovery that these defects are not as a result of poor RAD51 loading, as depletion of RAD51 itself does not give rise to the same RPA and fragmentation defect.

This is an important and potentially revealing finding. The paper itself is well written and I enjoyed reading it very much (something I rarely say in such reviews). With just a few exceptions the experiments are well controlled.

I have two major comments:

Firstly I feel that the central surprise, that there is a BRCA2 active role preventing NHEJ and MHMEJ that is not a consequence of failed RAD51 loading (that BRCA2 is responsible for) needs a little more evidence and elucidation. Currently the conclusion rests on RAD51/RAD51C depletion.

Experiments that might strengthen this finding include:

A. An add back of BRCA2 bearing mutations unable to load RAD51, as these would nevertheless be expected to restore RPA and chromosome stability.

B. Following the findings it should be possible to restoration of aspects of HR (RAD51 loading) while still observing the instability, for example by RAD51 over-expression, which can restore HR in BRCA2 deficient cells, or Dss1 over-expression.

The most obvious (to this reviewer) mechanism would appear to be RPA stabilization.

C. Could RPA over-expression restore stability?

The second major comment concerns timing – the proposed model implies that BRCA2, if directly inhibiting NHEJ –mechanisms, but recruit before or at the same time as 53BP1/RIF1/Artemis. Can the authors supply evidence that this is the case?

Minor comments:

Technically I find most of the experiments well controlled and convincing. However the claim in the abstract that Artemis requires RIF1 and 53BP1, while correct reads as though this applies especially to BRCA2 deficient cells. Experiment fig 7f would be informative if also undertaken in BRCA2 deficient cells.

The materials and methods should say which human cells were used.

Point-by-point responses to the reviewers:

We would like to thank both reviewers for their insightful comments. We have now carried out additional experimentations as advised, and hope that results from these experimentations adequately address the reviewers' concerns and provide further support to the main conclusions. Below is our point-by-point response to the reviewers' comments:

Reviewers' comments:

Reviewer #1 (Remarks to the Author):

It has also previously been shown that BRCA2 has a RAD51-independent role in blocking stalled replication fork degradation by MRE11 (see comment below). This latter role is less well characterised and is the focus of this paper. Here it is shown that in the absence of BRCA2 the broken DNA is rejoined by MMEJ and c-NHEJ (and not by RAD52-dependent SSA as might be expected) with a requirement for Artemis and promoted by 53BP1/RIF1. These represent important additions to our understanding of how mis-rejoining occurs when DNA is nucleolytically degraded at the fork and generally I find the work interesting and novel.

Thanks for the nice summary!

I have two major comments – firstly the paper does not discuss the known literature adequately. Secondly, the nature of MMEJ is not sufficiently clear and an additional experiment could add clarity. These are given in points below.

1. There have been several papers reporting the role of BRCA2 (distinct from HR) in fork protection from MRE11. The first paper was by Schlacher and Jasin (PMID:21565612). There are several additional papers since then.eg PMC4973925 by Nuzzensweig (which reports a role for PTIP in this process.) These papers must be cited. This does not negate the work here, which rather describes the downstream consequences of fork degradation. Figures 1 are needed but how they overlap with known findings should be stated. This aspect is not novel and should not be presented as such. Thus, the entire first part of the paper should be rewritten. Additionally, the paper by Nussensweig describes a role for PTIP in this process, which is distinct to what is found here. These differences should be discussed. (PMC4614542 FANCD2, FANCI and BRCA2 cooperate to promote replication fork recovery independent of the FA core complex). The novelty of this work should be clearly stated – ie defining how the degraded DNA is rejoined.

Thanks for your suggestion. We have now described the role of BRCA2 in fork protection in the Introduction section and have cited these papers (Please see References 40-45).

The reviewer is correct in pointing out that PTIP contributes to the degradation of nascent DNA in BRCA2-deficient cells by recruiting the Mre11 nuclease to stalled replication forks (Ray Chaudhuri et al., 2016, Nature). Notably, the function of

PTIP at stalled replication forks is distinct from its DSB-dependent interactions with 53BP1 and RIF1 (Ray Chaudhuri et al., 2016, Nature), where BRCA2 functions during perturbed replication in a HR- and DSB-independent manner. In our present study, we emphasize on the consequence of BRCA2 loss following DSB induction. Our results suggest that BRCA2 may antagonize the 53BP1-RIF1-Artemis axis at extensively resected DSBs to inhibit alt-NHEJ and SSA (Please see Figure 8). Our findings in the present study uncovered a HR-independent but DSB-dependent function for BRCA2. Based on these findings, we speculate that, in addition to its essential role in HR, BRCA2 has at least two HR-independent functions: (1) it protects nascent DNA strands from PTIP/Mre11-mediated degradation at stalled replication forks (in a DSB-independent manner); (2) it protects the long ssDNA generated by extensive resection from 53BP1/RIF1/Artemis-mediated degradation at DSB sites (in a DSB-dependent manner). We have now included these information in the revised Discussion section.

2. In the literature MMEJ is often taken to represent Alt-NHEJ (as stated here). However, a recent study by the Lohr laboratory (Biehs et al; PMID: 28132842) and others have shown that c-NHEJ can also rejoin using small microhomology (MH, which I believe is a now recognized concept. Generally, Alt-NHEJ uses longer regions of MH (still shorter than HR) whereas c-NHEJ uses shorter MH but the precise overlap is unclear - whether c-NHEJ can use 9bp (as occurs in the reporter assay) is unclear. As discussed in the Biehs et al paper, MMEJ represents rejoining involving MH – it can arise via Alt-NHEJ, c-NHEJ or RAD52-dependent SSA. Thus, distinguishing MMEJ and c-NHEJ is misleading – the more meaningful distinction is Alt-NHEJ and c-NHEJ. The requirement for Pol theta and the additivity of pol theta/XRCC4 loss is the strongest evidence that the process reflects Alt-NHEJ (Artemis appears to have functions in both c-NHEJ and Alt-NHEJ). However, whether pol theta actually functions in some c-NHEJ has not been well examined, since this process is poorly understood. The most informative analysis would be to examine the ligase required since this distinguishes the processes. Alt-NHEJ is ligase I/III dependent whereas c-NHEJ is ligase IV dependent. A partial requirement for ligase IV is already shown. Thus, it would be beneficial to examine siRNA of ligase I/III and/or PARP inhibition. I do not feel KO of ligase I/III is required (ie. siRNA should suffice) (nor dissection of the individual contribution of ligase I/III) but merely the demonstration that these ligases contribute especially when ligase IV is absent (and that there is complete loss when both are absent). The analysis of RAD52 argues that the rejoining process is not SSA. This would then allow the conclusion that the rejoining processes are Alt-NHEJ and c-NHEJ with MH usage and Artemis-dependency. The MH usage could be discussed briefly. This avoids the conclusion (as currently presented) that c-NHEJ does not involve MH – which is not known. Biehs et al. It would be useful to have this put in context in the discussion.

Thanks for the clarification and we have now used alt-NHEJ instead of MMEJ in the revised manuscript to avoid any confusion in the field. In addition, we have now discussed the usage of MH in resection-dependent c-NHEJ in G1 cells (Biehs et al., 2017, Molecular cell) in the revised Discussion section.

According to this reviewer's suggestion, we have now examined whether downregulation of ligase I/III would be able to suppress the gross genomic instability caused by BRCA2 inactivation. As shown in the revised Supplementary Figure 4a-e, like Polθ, depletion of ligases I/III partially suppressed the accelerated dissolution of RPA2 foci and nuclear fragmentation in BRCA2-depleted cells, further supporting the idea that, in addition to c-NHEJ, alt-NHEJ also contributes to the gross genomic instability observed in BRCA2-deficient cells. More importantly, downregulation of ligases I/III in ligase IV-deficient cells almost completely suppressed the observed defects induced by BRCA2 depletion (Please see the revised Supplementary Figure 4a-e). These data are consistent with our original hypothesis that both alt-NHEJ and c-NHEJ pathways contribute to the gross genomic instability observed in BRCA2-deficient cells.

3. Extending from this, I am somewhat confused by the finding the nuclear fragmentation is abolished when LigIV/XRCC4 are absent. If c-NHEJ has a function, and if RPA foci remain, surely there should be more nuclear fragmentation – maybe not aberrantly rejoined fragments (ie rearrangements) but certainly unrejoined fragments. This is confusing and should be discussed – can unrejoined fragments be distinguished from rearrangements? The figure presented appears to show many unrejoined fragments - so why are they lost when XRCC4/LigIV is absent?

We apologize for the lack of clarity. We proposed that, if the long ssDNA overhangs generated by extensive resection in BRCA2-deficient cells does not contain short regions of homology, Artemis would process the long 3' ssDNA overhangs to generate substrates suitable for c-NHEJ. Thus, in the presence of LigIV/XRCC4, these DSBs in BRCA2-depleted HR-compromised cells are diverted to repair via the c-NHEJ pathways, resulting in large deletions at DSB sites, chromosome missegregation, and nuclear fragmentation (Figure 1g and Supplementary Movies 1-2). By contrast, in the absence of LigIV/XRCC4, these DSBs in BRCA2-depleted HR-compromised cells are not able to be repaired via c-NHEJ, resulting in unrejoined fragments and persistent DSBs. The presence of persistent DSBs results in cell cycle arrest in the G2 phase and eventually cell death (in this case, no massive nuclear fragmentation will be observed). Thus, loss of DNA ligase 4 or XRCC4 partially suppresses the heightened frequency of nuclear fragmentation in BRCA2-depleted cells.

4. The role for 53BP1 here is surprising and distinct to other roles of 53BP1, which is to protect against resection – here actually loss of 53BP1 prevents degradation (MRE11-dependent) ie RPA foci loss. Indeed, 53BP1 is often described as being pro-NHEJ whereas in this case it appear to protect against NHEJ usage. This could be usefully brought out in the discussion.

We apologize for the lack of clarity. We proposed that, if resection in BRCA2-deficient cells fails to expose any DNA stretches with microhomologies, 53BP1-RIF1-Artemis would process the long 3' ssDNA overhangs to generate

substrates suitable for c-NHEJ (Please see Figure 8). Thus, in this case 53BP1 also promotes c-NHEJ.

Reviewer #2 (Remarks to the Author):

This paper describes the consequence of BRCA2 loss. Surprisingly this leads to a reduction in RPA foci, the exploration in the manuscript as to the pathways involved reveals NHEJ and MMEJ as those responsible. (As might be expected SSA is not). Moreover BRCA2 depleted cells exhibit severe chromosome fragmentation that is also a result of MMEJ and NHEJ usage. They reveal that ARTEMIS in particular plays a role.

The most surprising insight for me comes in the discovery that these defects are not as a result of poor RAD51 loading, as depletion of RAD51 itself does not give rise to the same RPA and fragmentation defect. This is an important and potentially revealing finding. The paper itself is well written and I enjoyed reading it very much (something I rarely say in such reviews). With just a few exceptions the experiments are well controlled.

Thanks for the nice summary!

I have two major comments:

Firstly I feel that the central surprise, that there is a BRCA2 active role preventing NHEJ and MHMEJ that is not a consequence of failed RAD51 loading (that BRCA2 is responsible for) needs a little more evidence and elucidation. Currently the conclusion rests on RAD51/RAD51C depletion.

Experiments that might strengthen this finding include:

A. An add back of BRCA2 bearing mutations unable to load RAD51, as these would nevertheless be expected to restore RPA and chromosome stability.

Thanks for your suggestion. Since the BRCA2 BRC1, -2, -3, and -4 bound to RAD51 with higher affinity than either BRC5, -6, -7, or -8 (Carreira et al., Proceedings of the National Academy of Sciences of the United States of America, 2011), we generated a BRCA2 internal deletion mutant that lacks the first four BRC repeats (Δ -BRC1-4). However, the Δ -BRC1-4 mutant was still able to restore RAD51 foci formation in BRCA2-depleted cells, albeit to a lesser extent than wild-type BRCA2 (Data not shown). These findings suggest that the eight BRC repeats in BRCA2 are redundant for RAD51 recruitment to sites of DNA damage.

Considering that deletion of the whole central region of BRCA2 (containing all eight BRC repeats) may severely disrupt the protein structure and function, we decided to use a wild-type and a mutated BRC4 repeat of BRCA2 (Chen et al., 1999, The Journal of Biological Chemistry). As shown in the revised Figure 2d-f, whereas overexpression of a wild-type, but not a mutated, BRC4 repeat of BRCA2 significantly impaired RAD51 foci formation as previously reported (Chen et al.,

1999, *The Journal of Biological Chemistry*), it had no significant effect on the kinetics of the dissolution of RPA2 foci or on nuclear integrity. These results further strengthen the idea that BRCA2 suppresses gross genomic instability independently of RAD51.

B. Following the findings it should be possible to restoration of aspects of HR (RAD51 loading) while still observing the instability, for example by RAD51 over-expression, which can restore HR in BRCA2 deficient cells, or Dss1 over-expression.

Thanks for your suggestion. We have now examined whether overexpression of RAD51 may suppress the gross genomic instability phenotype caused by BRCA2 inactivation. As shown in the revised Supplementary Figure 2a-c, overexpression of RAD51 was unable to stabilize RPA2 foci or suppress the increased frequency of nuclear fragmentation. These findings are consistent with our original hypothesis in which BRCA2 suppresses gross genomic instability independently of RAD51.

The most obvious (to this reviewer) mechanism would appear to be RPA stabilization.

C. Could RPA over-expression restore stability?

Thanks for your suggestion. We have now examined whether overexpression of RPA may suppress the gross genomic instability phenotype caused by BRCA2 inactivation. As shown in the revised Supplementary Figure 7a-c, overexpression of RPA was unable to stabilize RPA2 foci or suppress the increased frequency of nuclear fragmentation in BRCA2-depleted cells. These results suggest that, even in the presence of excess RPA, Pol θ and/or unknown proteins are still able to release RPA from ssDNAs in BRCA2-depleted cells to promote MMEJ. These findings are consistent with our original hypothesis that BRCA2 antagonizes Pol θ /unknown protein-mediated RPA release from ssDNAs during MMEJ.

The second major comment concerns timing – the proposed model implies that BRCA2, if directly inhibiting NHEJ –mechanisms, but recruit before or at the same time as 53BP1/RIF1/Artemis. Can the authors supply evidence that this is the case?

As suggested by this Reviewer, we have now performed the laser micro-irradiation experiments. As shown in the revised Supplementary Figure 6, BRCA2 accumulated at sites of laser-inflicted DNA damage tracks with kinetics similar to 53BP1.

Minor comments:

Technically I find most of the experiments well controlled and convincing. However the claim in the abstract that Artemis requires RIF1 and 53BP1, while correct reads as though this applies especially to BRCA2 deficient cells. Experiment fig 7f would be informative if also undertaken in BRCA2 deficient cells.

Thanks for your suggestion. We have now performed the micro-irradiation experiment in both wild-type and BRCA2-depleted HeLa cells. As shown in the

revised Figure 7f-g, downregulation of 53BP1 or RIF1 in both wild-type and BRCA2-depleted cells hampered the retention of Artemis at laser-generated DSBs.

The materials and methods should say which human cells were used.

As suggested, we have now provided this information in the Materials and Methods section.

References:

Biehs R, *et al.* DNA Double-Strand Break Resection Occurs during Non-homologous End Joining in G1 but Is Distinct from Resection during Homologous Recombination. *Molecular cell* **65**, 671-684 e675 (2017).

Carreira A, Kowalczykowski SC. Two classes of BRC repeats in BRCA2 promote RAD51 nucleoprotein filament function by distinct mechanisms. *Proceedings of the National Academy of Sciences of the United States of America* **108**, 10448-10453 (2011).

Chen CF, Chen PL, Zhong Q, Sharp ZD, Lee WH. Expression of BRC repeats in breast cancer cells disrupts the BRCA2-Rad51 complex and leads to radiation hypersensitivity and loss of G(2)/M checkpoint control. *The Journal of biological chemistry* **274**, 32931-32935 (1999).

Ray Chaudhuri A, *et al.* Replication fork stability confers chemoresistance in BRCA-deficient cells. *Nature* **535**, 382-387 (2016).

REVIEWERS' COMMENTS:

Reviewer #1 (Remarks to the Author):

The authors have done a good job in addressing my concerns as well as those of the other reviewer. I am supportive of acceptance and believe this is an interesting study. I have just one comment that would be good for the authors to consider before acceptance. The changes required are only writing and not experimental.

Throughout the paper the authors consider the role of BRCA2 in HR arising at a two ended DSB generated by IR. However, HR plays a relatively minor role in repairing directly induced DSBs with the major role being in promoting replication fork recovery after fork stalling/collapse - this may well involve a one-ended DSB.

the model presented in Figure 8 creates a problem -- effectively the authors argue that there is gross rearrangements seen when BRCA2 is depleted due to rejoining by c-NHEJ or Alt NHEJ after resection. However, such rejoining (and as depicted in the figure) should give deletions at the junctions but not massively increased rearrangement (which comes from rejoining the wrong DNA ends). This might occur to some extent but not as massively as described (and not depicted in the figure). I think the discrepancy /inconsistency is that most of the DSBs being handled by BRCA2 probably arising the fork and are probably one-ended DSBs. Hence, to have c-NHEJ rejoining them has to be toxic. I believe there would be much more clarity if this was discussed and/or shown in the figure.

I apologise for not mentioning this in the previous report.

Reviewer #2 (Remarks to the Author):

The authors have performed all the experiments that the reviewers requested and the results are extremely enlightening. The case for the proposed model is much stronger and I would like to support the manuscripts publication. The surprising data will be of interest to many in the DNA repair and replication fields.

Point-by-point responses to the reviewers:

Reviewers' comments:

Reviewer #1 (Remarks to the Author):

The authors have done a good job in addressing my concerns as well as those of the other reviewer. I am supportive of acceptance and believe this is an interesting study. I have just one comment that would be good for the authors to consider before acceptance. The changes required are only writing and not experimental.

Throughout the paper the authors consider the role of BRCA2 in HR arising at a two ended DSB generated by IR. However, HR plays a relatively minor role in repairing directly induced DSBs with the major role being in promoting replication fork recovery after fork stalling/collapse - this may well involve a one-ended DSB.

the model presented in Figure 8 creates a problem -- effectively the authors argue that there is gross rearrangements seen when BRCA2 is depleted due to rejoining by c-NHEJ or Alt NHEJ after resection. However, such rejoining (and as depicted in the figure) should give deletions at the junctions but not massively increased rearrangement (which comes from rejoining the wrong DNA ends). This might occur to some extent but not as massively as described (and not depicted in the figure). I think the discrepancy /inconsistency is that most of the DSBs being handled by BRCA2 probably arising the fork and are probably one-ended DSBs. Hence, to have c-NHEJ rejoining them has to be toxic. I believe there would be much more clarity if this was discussed and/or shown in the figure.

I apologies for not mentioning this in the previous report.

Thanks for your suggestion. We have now discussed this in the revised manuscript.

Reviewer #2 (Remarks to the Author):

The authors have performed all the experiments that the reviewers requested and the results are extremely enlightening. The case for the proposed model is much stronger and I would like to support the manuscripts publication. The surprising data will be of interest to many in the DNA repair and replication fields.

Thanks!